

# How the meteorological spectral nudging impacts on aerosol radiation clouds interactions?

Laurent MENUT[1], Bertrand BESSAGNET[1,2], Arineh CHOLAKIAN[1], Guillaume SIOUR[3], Sylvain MAILLER[1,4], and Romain PENNEL[1]

[1]Laboratoire de Météorologie Dynamique (LMD), Ecole Polytechnique, Institut Polytechnique de Paris, ENS, PSL Research University, Sorbonne Université, CNRS, Palaiseau, France
[2]Now at Joint Research Centre, European Commission, Ispra, Italy
[3]Univ Paris Est Créteil and Université Paris Cité, CNRS, LISA, 94010 Créteil, France
[4]Ecole des Ponts-ParisTech, Marne-la-Vallée, France

**Correspondence:** Laurent Menut, menut@lmd.ipsl.fr

**Abstract.**

Meteorological and chemical modelling at the regional scale involve nudging to remain consistent with large-scale meteorology and meteo-chemical coupling to properly consider the interactions between aerosols, clouds and radiations. Both types of process can change meteorology, but not for the same reasons and not necessarily in the same way. To assess the possible

interactions between nudging and coupling, several simulations are carried out with the WRF-CHIMERE coupled model. By comparison with measurements, we show that the use of nudging significantly improves the model performances. We also show that coupling changes the results much less than nudging. Finally, we show that when nudging is used, it limits the variability in the results due to coupling.

## 1 Introduction

The regional modelling of atmospheric pollution includes the modelling of meteorology and chemistry-transport. If the chemistry transport model (CTM) receives information from the meteorological model but does not send it back, it is an offline modeling. If, on the other hand, the two models exchange information, we are in an online modelling mode. Being regional, the two models need forcing at least at the boundaries of the domain, and possibly inside the domain. For the meteorological part, the grid or spectral nudging technique is used, (von Storch and Zwiers, 2001), (Kruse et al., 2022). With the spectral

nudging, the meteorology can evolve due to mesoscale turbulence, but large-scale atmospheric circulations remain consistent with the global modelling that serves as forcing. Knowing that the global model has been corrected by data assimilation, the meteorological fields already implicitly contain the effects of aerosols on meteorology, Figure 1.

On the other hand and for chemistry-transport modeling (CTM) and in online mode (increasingly used today), aerosols will modify the meteorology within the simulation domain. These changes are performed at higher spatial and temporal scales than

the forcing which is intrinsically a fine scale process. Above all, they are completely independent of the large-scale circulation.





It is therefore possible to have a contradiction between the scales: aerosols will modify the meteorology on the small scale, while at the same time, nudging will constrain the large scale to remain close to the initial global forcing.

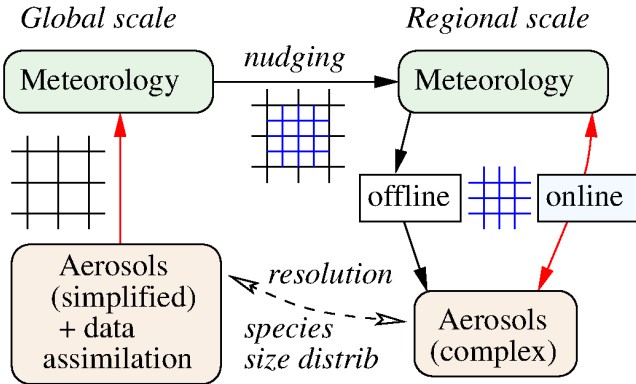

**Figure 1.** *The paradox of the regional model nudged by a global model using other aerosol forcing.*

The effect of nudging on the modelling of regional meteorology is paradoxical: nudging improves the realism of simulations by forcing them to stay close to the observed reality, but this is achieved by introducing unrealistic inconsistencies between

the dynamics and the physics of the model, therefore possibly limiting or distorting, (Lin et al., 2016), our understanding of processes by dampening the effect of model parameterizations. As presented in Figure 1, the global scale (used for nudging) and the regional scale are supposed to represent the same physical reality, but they rely on different aerosol forcings, spatial resolutions and parameterizations, therefore leading to divergent meteorology for the same location. We therefore have two processes acting in parallel: data assimilation on large-scale fields (global forcing, for example) and meteorological and

chemical-transport coupling at a smaller scale. This paradox leaves the modeller with a methodological alternative. Either avoid nudging and let model physics operate freely, ensuring consistency between the physics and the dynamics, or use nudging and ensure that the model stays close enough to observations, but at the cost of introducing inconsistencies between the dynamics and the physics.

This methodological alternative has already been reported in regional and global climate modelling, particularly discussing

the good use of nudging to evaluate model sensitivity to a forcing or to parameterization choices. It is already well-known in that field that the use of nudging techniques, while indispensable for the representation of individual events in a realistic way, can dampen the response of models to other effects such as air-sea coupling (Berthou et al., 2016) or convective parameterizations, (Song et al., 2011). While improving the representation of individual events, nudging forces a model to reproduce a large-scale variability not necessarily in equilibrium with its physical parameterizations, thereby introducing inconsistency between the

dynamics and physics, (Pohl and Crétat, 2013). On the bright side, nudging reduces the internal variability in the model, and therefore the spread between several different realizations, such as the sensitivity studies performed to evaluate the effect of a particular process or parameterization, permitting robust detection of such effects from shorter simulations, (Sun et al., 2019). For example, Kooperman et al. (2012) shows that, by attenuating the 'natural variability' between two sensitivity simulations,





nudging permits to isolate the direct effect of a physical process from natural variability. In summary, the effect of nudging on

sensitivity studies is twofold. On the one hand, it dampens the effect of a change in processes or parameterizations (Song et al., 2011; Pohl and Crétat, 2013; Lin et al., 2016; Berthou et al., 2016) and introduces inconsistencies between the dynamics and the physics (Pohl and Crétat, 2013; Lin et al., 2016), but on the other hand it strongly reduces the internal (chaotic) variability of meteorology in the numerical simulations, and thereby permits to observe sensitivity effects in a more robust way (Kooperman et al., 2012; Lin et al., 2016; Sun et al., 2019), even in relatively short simulations as it will be the case in the present study.

The effect of nudging on regional simulation has been studied mainly on meteorological variables such as temperature and precipitation. Using the WRF model, (Powers et al., 2017), Glisan et al. (2013) studied the effect of the nudging on arctic temperature and precipitation. They showed that the strength of the nudging is not a sensitive key for the results. Spero et al. (2014) proposed changes in the spectral nudging to improve clouds, radiation and precipitation in their WRF simulations. The study of (He et al., 2017) is on climatological time-scale and more specifically on the possible changes in temperature due to

the combined effects of large-scale forcing and regional aerosol/radiation interactions. They concluded that the use of nudging is possible and realistic for aerosol radiative effect studies, but with caution, the smaller the spatial scale. Rizza et al. (2020) explored the sensitivity of the WRF model to various configurations, including the spectral nudging. They compared their results to meteorological (wind, temperature) surface measurements and conclude there is no interest to nudge the meteorology inside the boundary layer.

For the impact of this methodology on pollutants concentrations, the focus of the present study, the studies are very rare. In these cases, they are more dedicated to regional climate (trends, long term scenarios) that to regional atmospheric pollution cases. One of the first study is (Hogrefe et al., 2015) which performed simulations tests in the framework of the AQMEII2 project and showed that the nudging reduces the bias for temperature with or without aerosol effects. They showed that, on temperature, the effect of the nudging is larger than the effect of the feedback of aerosols on meteorology. The same question

arises in (He et al., 2017) about the relative impact of temperature nudging compared to aerosol radiative effects. They showed that the nudging has less effect than the aerosol-radiation interactions at global and regional scale, but could be more important at the local scale.

In the present study we will focus at regional scale modelling and spectral nudging impact both on meteorology and pollutants concentrations for a limited temporal scale. Simulations of the same case are carried out to evaluate the weight of

nudging on cloud/radiative/aerosol interactions. The key question here is to know on a regional scale what is most important for pollutant concentrations between nudging and coupling and how they interplay. Even if they are not the same kind of processes and not directly comparable, they are often 'free parameters' up to the user, then it is important to well understand their relative weight on modelled pollutants surface concentrations. The Section 2 describes the models used and the simulations configurations. The Section 3 presents the results of various simulations performed with the WRF and CHIMERE models. The

Section 4 presents refined results in case of the online coupling. Finally, conclusions are presented.





## 2    The modelling system

The two models used in this study are WRF 3.7.1, (Powers et al., 2017), and CHIMERE 2020r3, (Menut et al., 2021). The simulations are done over a single domain, with an horizontal resolution of 50 km × 50 km, as presented in Figure 2. Simulations are performed from $1^{st}$ July to $31^{st}$ August 2022. It corresponds to the same domain and the same period as presented in

(Menut et al., 2023).

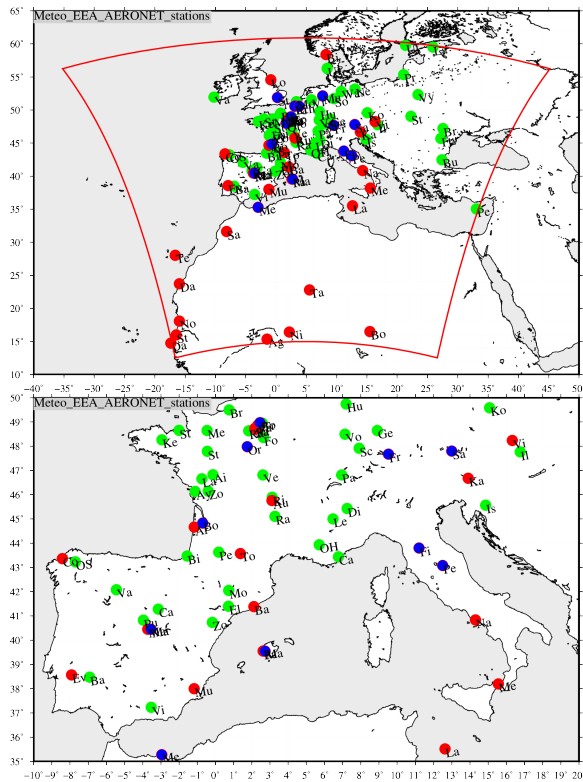

**Figure 2.** *Maps of measurements stations of meteorological stations (blue points), EEA (green points) and AERONET (red points). A zoom is presented over Western Europe, where stations are more numerous. For readability, only the two letters of each name is reported. The complete list of stations with their coordinates is presented in Table A1.*

The model configurations consist in several ways to take into account the spectral nudging in WRF. It also consists to take into account aerosols direct and indirect effects or not. This leads to four different simulations as explained in Table 1.





| Simulation | Nudging | Coupling | |
| --- | --- | --- | --- |
| no_nudg_offline | | | no nudging and offline modelling |
| no_nudg_online | | ✓ | no nudging and online modelling |
| nudg_offline | ✓ | | spectral nudging and offline modelling |
| nudg_online | ✓ | ✓ | spectral nudging and online modelling |

**Table 1.** *Simulations performed for this study*

## 2.1 The WRF model set-up

### 2.1.1 The main schemes used

Many physical schemes are available in WRF and many studies quantified the impact of several combinations on the results, (Cohen et al., 2015). The model is used with a constant horizontal resolution of 50 km × 50 km, on an horizontal grid of 103 × 106 cells, and 28 vertical levels from the surface to 50 hPa. The Single Moment-5 class microphysics scheme is used, allowing for mixed phase processes and super cooled water (Hong et al., 2004). The radiation scheme is RRTMG scheme with the MCICA method of random cloud overlap (Mlawer et al., 1997). The surface layer scheme is based on Monin-Obukhov

with Carslon-Boland viscous sub-layer. The surface physics is calculated using the Noah Land Surface Model scheme with four soil temperature and moisture layers, (Chen and Dudhia, 2001). The planetary boundary layer physics is processed using the Yonsei University scheme, (Hong et al., 2006) and the cumulus parameterization uses the ensemble scheme of (Grell and Dévényi, 2002).

### 2.1.2 The nudging choices

Several studies were devoted to the comparison between grid and spectral nudging, (Liu et al., 2012; Vincent and Hahmann, 2015; Ma et al., 2016; Zittis et al., 2018). The two approaches have strength and weaknesses. Grid nudging has the advantage to be applied over all grid cells, when spectral nudging is applied only in zonal and meridional directions and only for some predefined wavenumbers. Grid nudging seems more appropriate for precipitation intensity, (Ma et al., 2016). On the other hand, spectral nudging is less intrusive at small scale and give the regional model more freedom than the large-scale forcing.

One can also note that spectral nudging has a more important numerical cost than grid nudging, (Zittis et al., 2018). In this study, we prefer to use spectral nudging to have more variability at the regional scale.

Usually, the spectral nudging is applied to four meteorological variables: the wind components $u$ and $v$, the temperature and the perturbation of the geopotential height. The nudging of specific humidity is avoided, considered as badly represented at the largest scale by the forcing model, (Heikkila et al., 2010; Otte et al., 2012). The nudging of water vapor (or moisture) is also

avoided by Liu et al. (2012), considering that this variable has no large scale features as strong as the other meteorological fields. In addition, nudging at the same time temperature and humidity may produce inconsistencies, (Sun et al., 2019). However, Spero et al. (2014) consider that not nudging this variable may be at the origin of the overestimation of precipitation when





using spectral nudging, compared to grid nudging. They considered that nudging moisture guarantees to treat consistently thermodynamical fields (potential temperature and water mixing ratio). But this nudging should be restricted to below the tropopause due to too large values in the stratosphere with some global models. They also note that for moisture the best coefficient is $4.5 \times 10^{-5}$ s$^{-1}$ to be consistent with the input data fields having a 6 h frequency. Using a lower value ($3 \times 10^{-4}$ s$^{-1}$, then 1 h) induces an overprediction of precipitation. For all these studies, there is no nudging in the boundary layer.

An important parameter is the nudging coefficient (in s$^{-1}$), noted $g$. This coefficient may have different values, depending on the meteorological variable: the wind component $u$ and $v$ with $guv$, the temperature with $gt$ and the water vapor with $gq$. When using spectral nudging in WRF, it is also possible to nudge geopotential height perturbations $gph$. With the WRF model, the default value is equal to 0.0003 s$^{-1}$, corresponding to a value found in many studies such as Liu et al. (2012), Otte et al. (2012), Ma et al. (2016), Gomez and Miguez-Macho (2017), Zittis et al. (2018) and Huang et al. (2021). Some others studies made sensitivity experiments to quantify the impact of this value on the results such as Choi et al. (2009), Cha et al. (2011), Glisan et al. (2013), Spero et al. (2014), He et al. (2017) and Spero et al. (2018) and no significant impact was found. This value corresponds to a one hour frequency for the nudging use and is well representative of the large scale fields used as forcing as well as to the frequency of the data used for the analysis of the global fields. In this study, this value is used for all simulations with nudging. The wind components, the potential temperature perturbation and the water vapor mixing ratio are nudged using spectral nudging with a coefficient $g$=0.0003 s$^{-1}$. There is no nudging in the Planetary Boundary Layer (PBL).

The calculation frequency is set in order to have an active nudging every time-step. The wave numbers are calculated with the hypothesis that features greater than 1000 km are sufficiently well resolved in global model, (Gomez and Miguez-Macho, 2017). Then, the following equation is applied (for example for the $x$ direction):

$$x_{wn} = int\left(\frac{\Delta x \times N_x}{R}\right) \tag{1}$$

with $\Delta x$ the horizontal resolution (in meters), $N_x$ the number of grid cell and $R$ the Rossby radius value, (Silva and Camargo, 2018; Mai et al., 2020). For this study and the horizontal resolution of 50 km (both in zonal and meridian directions), this leads to a wavenumber of $x_{wn}$=5. The same value is found for $y_{wn}$, having the same grid size in the two directions and a close number of cells.

## 2.2 The CHIMERE model configuration

This v2020r3 version of CHIMERE is to date the last distributed one and is designed to take into account the direct effects of aerosols on cloud and radiation (the online mode) or not (the offline mode). The way these effects are taken into account is described in Briant et al. (2017) (for the direct effects) and Tuccella et al. (2019) (for the indirect effects). The model configuration is exactly the same than in Menut et al. (2023): it includes emissions from anthropogenic, biogenic, sea-salt, biomass burning, lightning NO$_x$ and mineral dust sources. It also includes gaseous and aerosol chemistry for tens of chemical species. For gases, the MELCHIOR 2 scheme is used as described in Menut et al. (2013) and Mailler et al. (2017). For aerosols, ten bins are used from 0.01 to 40 $\mu$m. Emissions include several contributions such as anthropogenic, biogenic, sea-salt,



dimethylsulfide, biomass burning, lightning $NO_x$ and mineral dust. The anthropogenic emissions are those of CAMS, (Granier et al., 2019). The dry deposition is modelled following the Zhang et al. (2001) scheme and the wet deposition following Wang et al. (2014). The biomass burning emissions are those of CAMS as described in (Kaiser et al., 2012) and with the additional term of burned area as presented in (Menut et al., 2022) and (Menut et al., 2023) and designed to calculate the impact of fires on additional mineral dust emissions, change of LAI (Leaf Area Index) and biogenic emissions.

## 2.3 The measurements data

For the surface pollutants concentrations, the European Environment Agency (EEA, https://www.eea.europa.eu) provides a full set of hourly data for particulate matter $PM_{2.5}$, $PM_{10}$, ozone ($O_3$) and nitrogen dioxide ($NO_2$) for a large number of stations in Western Europe. Only urban, rural and suburban background stations are used in this study considering that the industrial and traffic ones have an inadequate spatial representativity for the present model outputs. For the Aerosol Optical
Depth (AOD) and the Angström exponent (ANG), the *AErosol RObotic NETwork* (AERONET, https://aeronet.gsfc.nasa.gov/) level 1.5 measurements are used, (Holben et al., 2001). The AOD at a wavelength of $\lambda$=675 nm is daily averaged and compared to daily averaged modelled values. For 2m temperature and 10m wind speed, the measurements are provided by the Weather Information website of the University of Wyoming (UWYO) (http://www.weather.uwyo.edu/). Data are provided as integer values, restraining the accuracy of the comparison to the model results. The complete list of the measurements stations is
displayed in Table A1. Maps of the stations for which the measurements were used are presented in Figure 2.

## 3 Results

### 3.1 Statistical scores

For meteorological variables such as 2m temperature ($^oC$) and 10m wind speed (m.s$^{-1}$), measured by surface stations, statistical scores are presented in Table 2. These scores are calculated using all hourly data of the meteorological stations. For these
two variables, the best scores are obtained for the simulations with spectral nudging, but not systematically for the simulation with the coupling. For the temperature, the correlations are rather good. It is not the case for the wind speed, a more "local" variable more influenced by local features. It is logical, knowing the turbulent characteristics of the near-surface wind speed to have lower statistical scores than for temperature. In addition, the measurements are recorded in m.s$^{-1}$ as integer, biasing for low winds the comparison between model and measurements. But globally, it is noticeable that for meteorological variables,
the nudging configurations have always better statistical scores, logically these variables being nudged. The distinction between coupling or not is more subtle and the best statistical scores are more for the offline simulation, but with low differences with the online simulation.

For surface concentrations and optical properties, results are presented in Table 3 as statistical scores in order to quantify the relative impact of the coupling and of the spectral nudging. These scores are calculated by comparison between the modelled
outputs and the measured surface concentrations and optical properties, for the corresponding location and hour.





| Simulation | $R_s$ | $R_t$ | RMSE | bias |
|---|---|---|---|---|
| $T_{2m}$ | | | | |
| no_nudg_offline | 0.91 | 0.72 | 2.70 | -1.47 |
| no_nudg_online | 0.92 | 0.71 | 2.76 | -1.60 |
| nudg_offline | 0.93 | 0.77 | 2.21 | -1.24 |
| nudg_online | 0.93 | 0.78 | 2.27 | -1.34 |
| $u_{10m}$ | | | | |
| no_nudg_offline | 0.29 | 0.45 | 1.23 | 0.63 |
| no_nudg_online | 0.25 | 0.48 | 1.23 | 0.62 |
| nudg_offline | 0.38 | 0.57 | 1.03 | 0.48 |
| nudg_online | 0.38 | 0.55 | 1.07 | 0.52 |

**Table 2.** *Statistical scores for 2m temperature (K) and 10m wind speed (m.s$^{-1}$). Scores are aggregated for all stations and the spatial correlation is added to the temporal correlation. Calculations are done over the entire modelled period (July and August 2022). The best scores values are framed.*

For the surface concentrations, the three modelled chemical concentrations are compared against measurements, ozone, $PM_{2.5}$ and $PM_{10}$. The spatial correlation is always the same or better when the nudging is used. In case of nudging, the spatial correlation is more or less the same for the simulation with or without coupling. For the temporal correlation, the same type of result is observed: statistical scores are systematically better with the nudging. The impact of the coupling is less important and the scores are more or less the less with and without the coupling. The RMSE is systematically lower with the nudging as well as the bias, for the three variables.

For the optical properties, the conclusion are close that for the surface concentrations. The statistical scores are systematically better with the nudging than without. It is true for the AOD and the Angström exponent. The spatial and temporal correlations are better and the bias and the RMSE are reduced. As for surface concentrations, there is no clear impact of the use of the direct effects or not on the scores. The correlations are better in case of no nudging but less good with nudging. However, the impact remains low.

The conclusion of these results is that the differences between the simulations in the case of coupling (or not) is not significant. But, the differences in the case of nudging or not are significantly different and the use of the nudging always improves the simulations scores for all variables, the spatial and temporal correlations, the bias and the RMSE.

## 3.2 Time series of meteorological variables

As the study is based on nudging and coupling, it is important to compare impacts of the several simulations configurations on the meteorological variables. Simulations results are compared with surface measurements from meteorological stations in Europe and Africa. A list of the stations used is displayed in Table A1. For two stations in France, Orléans and Bordeaux,





| Simulation | $R_s$ | $R_t$ | RMSE | bias |
|---|---|---|---|---|
| **Ozone** | | | | |
| no_nudg_offline | 0.42 | 0.53 | 20.21 | -4.39 |
| no_nudg_online | 0.42 | 0.54 | 20.15 | -4.99 |
| nudg_offline | 0.45 | 0.63 | 18.11 | -2.51 |
| nudg_online | 0.45 | 0.62 | 18.18 | -2.91 |
| **PM$_{2.5}$** | | | | |
| no_nudg_offline | 0.12 | 0.40 | 4.18 | 1.35 |
| no_nudg_online | 0.10 | 0.41 | 4.30 | 1.46 |
| nudg_offline | 0.11 | 0.51 | 3.82 | 1.17 |
| nudg_online | 0.12 | 0.51 | 3.76 | 1.08 |
| **PM$_{10}$** | | | | |
| no_nudg_offline | 0.25 | 0.29 | 9.20 | -4.65 |
| no_nudg_online | 0.24 | 0.26 | 9.12 | -4.70 |
| nudg_offline | 0.25 | 0.37 | 8.80 | -5.39 |
| nudg_online | 0.27 | 0.37 | 8.93 | -5.43 |
| **AOD** | | | | |
| no_nudg_offline | 0.82 | 0.40 | 0.16 | -0.10 |
| no_nudg_online | 0.86 | 0.37 | 0.17 | -0.10 |
| nudg_offline | 0.88 | 0.54 | 0.16 | -0.10 |
| nudg_online | 0.86 | 0.52 | 0.16 | -0.10 |
| **Angstrom** | | | | |
| no_nudg_offline | 0.86 | 0.43 | 0.45 | -0.19 |
| no_nudg_online | 0.88 | 0.41 | 0.44 | -0.17 |
| nudg_offline | 0.91 | 0.54 | 0.36 | -0.09 |
| nudg_online | 0.90 | 0.52 | 0.37 | -0.08 |

**Table 3.** *Statistical scores for the surface ozone, PM$_{2.5}$, PM$_{10}$ ($\mu g.m^{-3}$) concentrations, AOD (no dim.) and Angstrom exponent (no dim.) by comparison with EEA and AERONET measurements and the four simulations. Scores are aggregated for all stations and the spatial correlation is added to the temporal correlation. Calculations are done over the entire modelled period (July and August 2022). The best scores values are framed.*

timeseries of daily averaged 2m temperature and 10m wind speed are presented in Figure 3. Note that this type of comparison
was also made for many others stations and the results are of the same kind. For the 2m temperature, simulations results are
close to the measurements during the whole period. The simulations are grouped into two sets: with and without nudging. The
simulations with and without coupling being very close. In Orleans and around the $20^{th}$ July, one can note that lower values are
correctly modelled by the 'nudging' simulations when the 'no_nudging' simulations overestimate the values. Other differences



are noted during the period $10^{th}$ to $20^{th}$ August when the simulations are different, both for the temperature and the wind

195 speed.

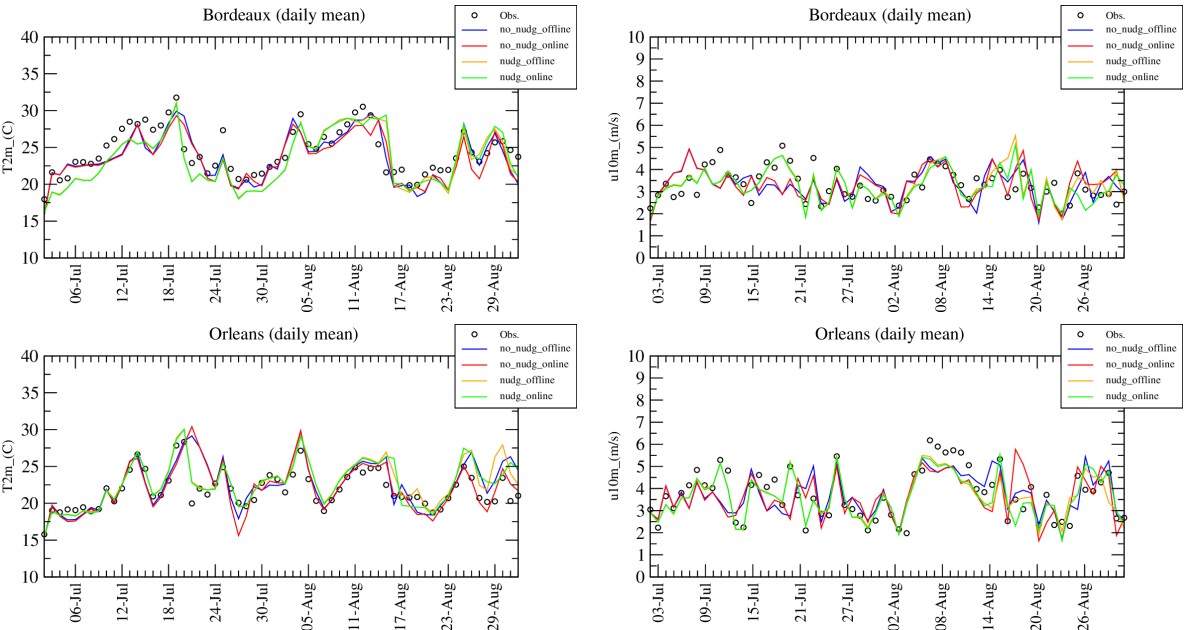

**Figure 3.** *Time series of daily mean 2m temperature ($^{o}C$) and 10m wind speed (m.s$^{-1}$) for the stations of Bordeaux and Orléans over the months of July and August 2022*

To better discuss these differences observed in August, a temporal zoom is done from $13^{th}$ to $20^{th}$ August and results are displayed in Figure 4. Data and model outputs are now presented at an hourly frequency. For the temperature, the first three days show a large durnal cycle with values ranging from 18 to 40 $^{o}$C, contrarily to the last three days with a reduced diurnal cycle between 16 and 25 $^{o}$C. The model is able to follow this weather change except at the interface and during the day of $15^{th}$

200 August when the model continue to have a large diurnal cycle not observed. Except for the $15^{th}$ August in Bordeaux and $16^{th}$ August in Orleans, the four simulations provides close values of temperature. It is not the case for the 10m wind speed where all four simulations provide very different values for the six consecutive days. For example, in Orleans, the variability of the wind speed is important, ranging from 1 to 10 m.s$^{-1}$, depending on the simulation configuration. There is no evidence as to which simulation best reproduces the observations, but the statistical scores (Table 2) show that the simulations with nudging

205 performs better.

### 3.3 Time series of surface concentrations

In order to have a more precise look at the results, time-series of surface concentrations of ozone and PM$_{2.5}$ in $\mu$g.m$^{-3}$, are presented in Figure 5. Results are presented for two sites, Biarritz and Fontainebleau. As already discussed in (Menut et al.,

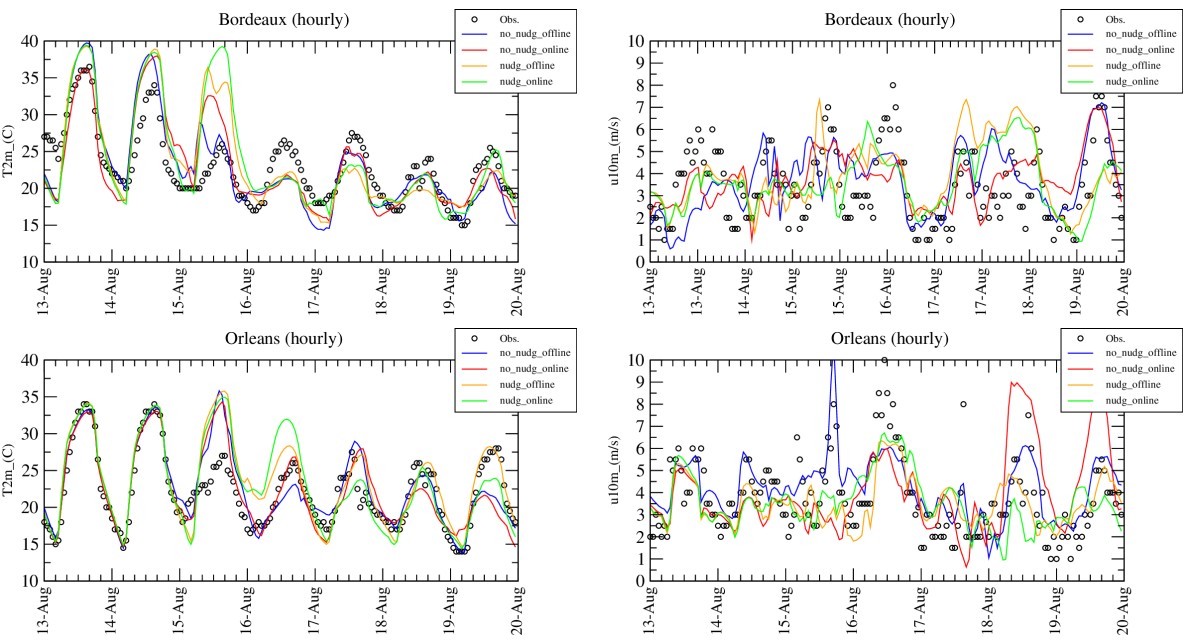

**Figure 4.** *Time series of hourly 2m temperature ($^oC$) and 10m wind speed ($m.s^{-1}$) for the stations of Bordeaux and Orléans and for the period 13 to 20 August 2022.*

2023), Biarritz, located in the South of France, was under the plumes of biomass burning coming from Spain mi-July and
mi-August 2022. Fontainebleau, near Paris, is not close to the Landes fires but was under their plume between the $12^{th}$ and
$18^{th}$ July 2022.

In Biarritz, two main peaks are recorded at the same time for ozone and $PM_{2.5}$: $16^{th}$ July and $14^{th}$ August 2022. For the
four simulations, the magnitude and the variability of the modelled concentrations are realistic and comparable to the surface
observations. It is difficult to disentangle the several simulations and to diagnose the best scores without statistical calculations.
The time-series exhibits an important day to day variability for all simulations. But a third peak is modelled but not measured
on July $30^{th}$. It is modelled only for the configurations without nudging. With nudging, the model removes this peak and is
then closer to the observations.

In Fontainebleau, the time variability is not the same between ozone and $PM_{10}$. But for all simulations, the model is close
to the observations and the day to day variability is well reproduced. For ozone, the largest differences are for the simulation
nudg_online with largest values on July $13^{th}$ and $19^{th}$, better corresponding to the measurements. For the same simulation,
and on July $21^{st}$, values are lower for $PM_{10}$ concentrations then closer to the observations than the other simulations. Note
that a peak is observed for ozone around the August $13^{th}$ but is not modelled by any of the four simulations. This probably
corresponds to long-range transport and an error in synoptic flow, and hence long-range ozone transport. This is because the

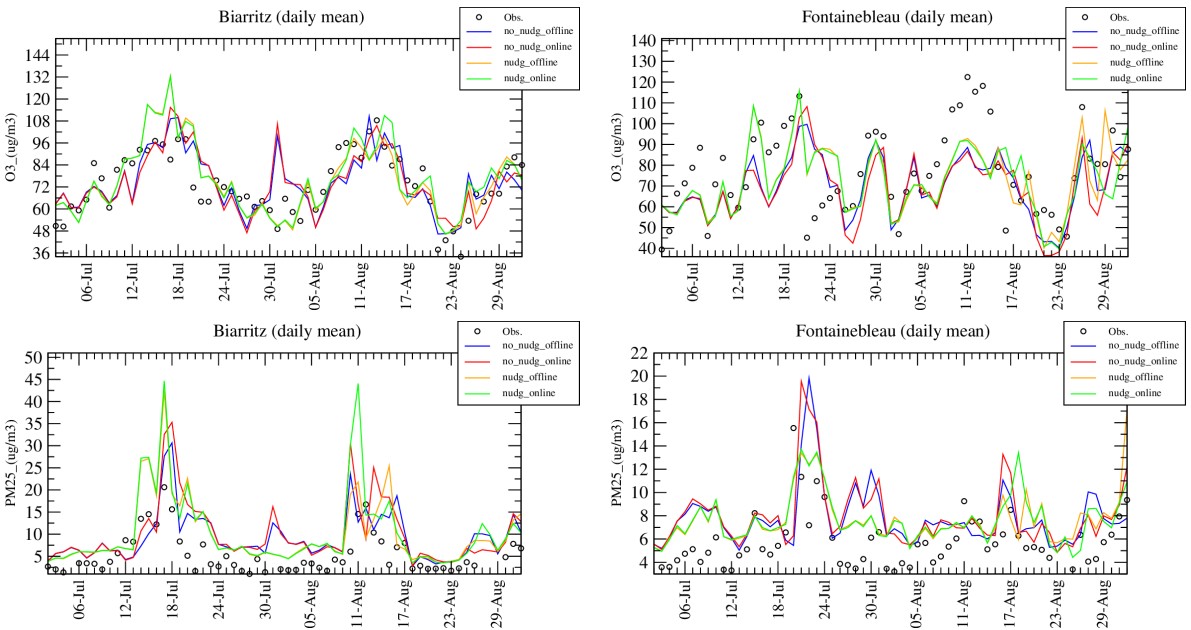

**Figure 5.** *Daily mean surface concentrations (in $\mu g.m^{-3}$) in Biarritz (left) and Fontainebleau (right) for ozone ($O_3$) and Particulate Matter with a mean mass median diameter less than 2.5 $\mu m$ ($PM_{2.5}$).*

configurations tested correspond to meteorological perturbations on a regional scale and within the study area. The fact that none of the four configurations simulates this peak shows that it is not due to a local or regional event.

In conclusion, the model simulations with nudging enable to avoid some non-observed peaks (such as ozone in Biarritz and $PM_{10}$ in Fontainebleau). The four simulations have all a large day to day variability, and there is no systematic bias between the simulations which could indicate a persistent effect of a process. The four simulations are comparable to the observations, there is no configuration being very false. It means that the use of nudging and coupling are not mutually exclusive, and that using spectral nudging, out of the boundary layer, probably does not interfere with coupling, which has more a local/regional effect.

### 3.4 Time series of optical depth

Time-series of daily mean values are presented in Figure 6 for Angström exponent (ANG) and Aerosol Optical Depth (AOD) in Birkenes, Barcelona and Toulouse. We can expect greater differences between simulations than for surface concentrations. ANG and AOD variables incorporate changes throughout the simulated atmospheric column, the troposphere. Therefore we take into account more possible changes between simulations, including changes on larger spatial scales such as aerosols long-range transport.



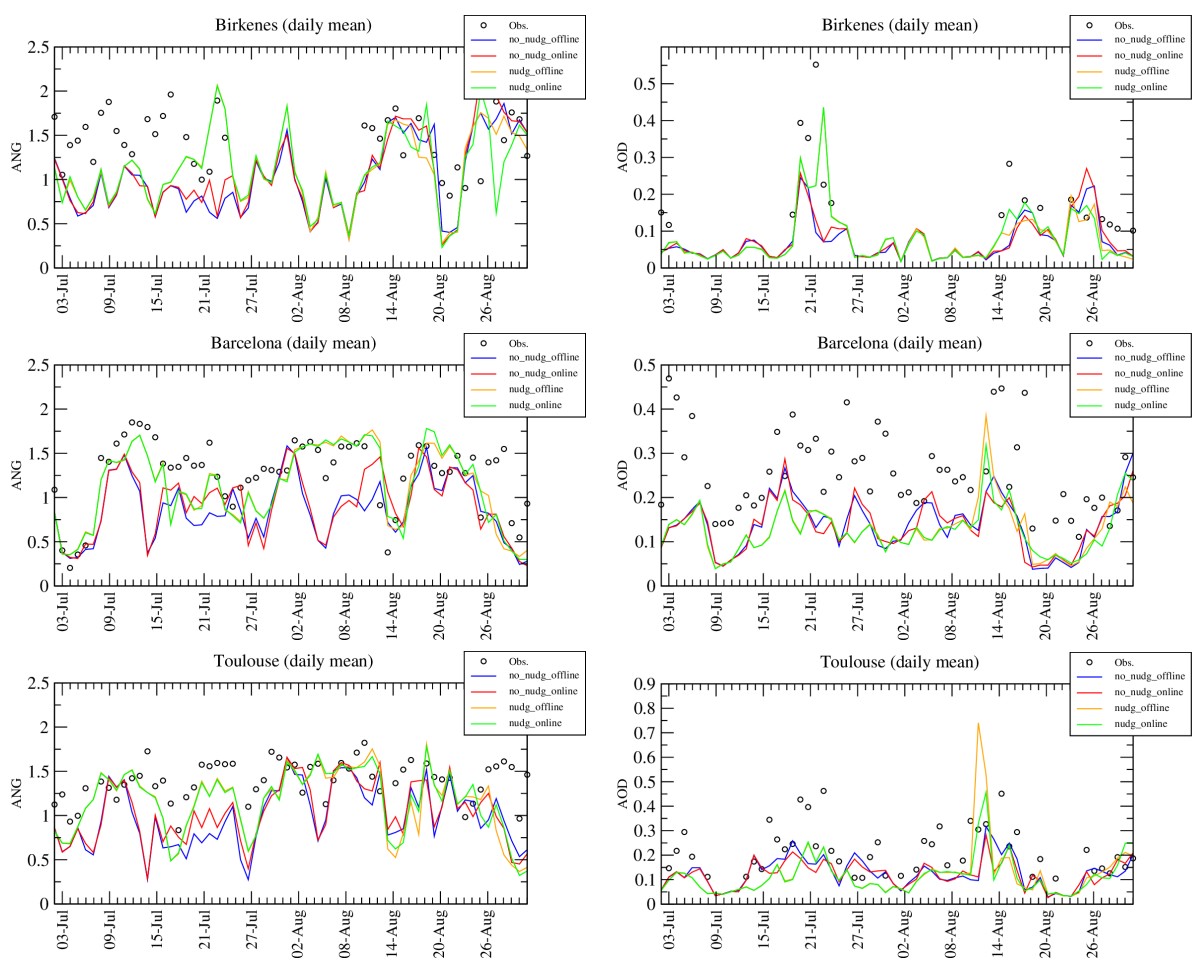

**Figure 6.** *Time-series of Ångström exponent (ANG, left) and Aerosol Optical Depth (AOD, right) in Birkenes, Barcelona and Toulouse.*

For the three sites and the two variables, one observes the same that for the surface concentrations. The two simulations with no nudging are close and the two simulations with nudging are also close. But the simulations with no nudging are very different that the simulations with nudging. This means that the direct/indirect effects are less dominant than nudging.

In Birkenes, the ANG time-series shows that the model overestimates the aerosols size by simulating coarse mode aerosols (low values of ANG) when the measurements are between 1.5 and 2 and representative of fine mode. This bias is mitigated by simulations with nudging, which better simulate the fine mode around July $21^{st}$. For this day and for AOD, the simulations with nudging also better simulate an important peak with observed values around 0.5. For the rest of the period, the four simulations are relatively close, both for ANG and AOD.

In Barcelona, the best capacity of the model to retrieve the observed ANG and AOD is for the simulations with nudging. It is clear for the period of July from $9^{th}$ to $16^{th}$ July when only the nudged simulations are able to simulate the high ANG





values. The same behaviour is observed during the period from $2^{sd}$ to $10^{th}$ August when the model correctly calculated ANG
around 1.5 when the non nudged simulations calculates low values (between 0.5 and 1). One can note that the simulation

without nudging show non negligible differences between them. For the AOD, the four simulations underestimate the values
compared to the observations. On $13^{th}$ August the two configurations with nudging are able to simulate an observed peak of
AOD contrarily to the simulations without nudging.

In Toulouse, the ANG values are between 1 and 1.5 showing relatively small particles. The day to day variability is close to
the one of Barcelona, with the same peaks at the same periods. All model configurations are close except for the period 13 to 19

July and for ANG: only the configurations with nudging are able to simulate high values of ANG close to the measurements.
For AOD, also only the nudged configurations are able to reproduce a peak representative of the measurements.

In conclusion, simulations with nudging give consistently better results, especially when ANG or AOD peaks are observed.
Differences can be seen between simulations without nudging and those with or without coupling. For simulations with nudg-
ing, there are no real differences for simulations with or without coupling. So we can see that nudging gives better scores but

leaves less variability than for the coupled configurations.

### 3.5   Time-averaged maps

Time-averaged maps are presented in this section. The averaged period is one month from $1^{st}$ to $31^{st}$ August 2022. Then,
differences are calculated between these averaged maps. Having four different configurations, the following differences are
calculated:

• (nonudg_online-nonudg_offline): impact of the coupling in case of no spectral nudging

• (nudg_online-nudg_offline): impact of the coupling in case of spectral nudging

• (nudg_offline-nonudg_offline): impact of the nudging with no direct/indirect effects

• (nudg_online-nonudg_online): impact of the nudging with direct/indirect effects

Results are presented in Figure 7 for the water vapor mixing ratio, this variable is particularly important for the radiative

transfer particularly at night. First of all, we note that the spatial structures of the difference values differ between the four
figures. The top figures show the impact of coupling, while the bottom figures show the impact of nudging. Depending on the
location, the differences may negative or positive. Large spatial structures exist showing that changes may affect large areas
or may be transported. There is no systematic location for the negative or positive changes. The impact of the coupling is
less important than the impact of nudging. The spatial structures are negative or positive and are not linked to vegetation or

mountainous areas or urbanized areas. The positive changes are more important than the negatives ones.

The same difference calculations are done for surface ozone concentrations. Figure 8a shows the effect of coupling (offline vs
online) in the case of spectral nudging. The differences are more significant and positive over North Africa with a maximum of
$+3\ \mu g.m^{-3}$. Over Western Europe the differences alternate between negative and positive values, but never exceed $\pm 1\ \mu g.m^{-3}$.
The non-zero differences are spatially very limited and the majority of the differences are below the low value of $\pm 0.4\ \mu g.m^{-3}$.

Figure 8b shows much larger differences over the whole simulation domain. Positive and negative differences can be over sea

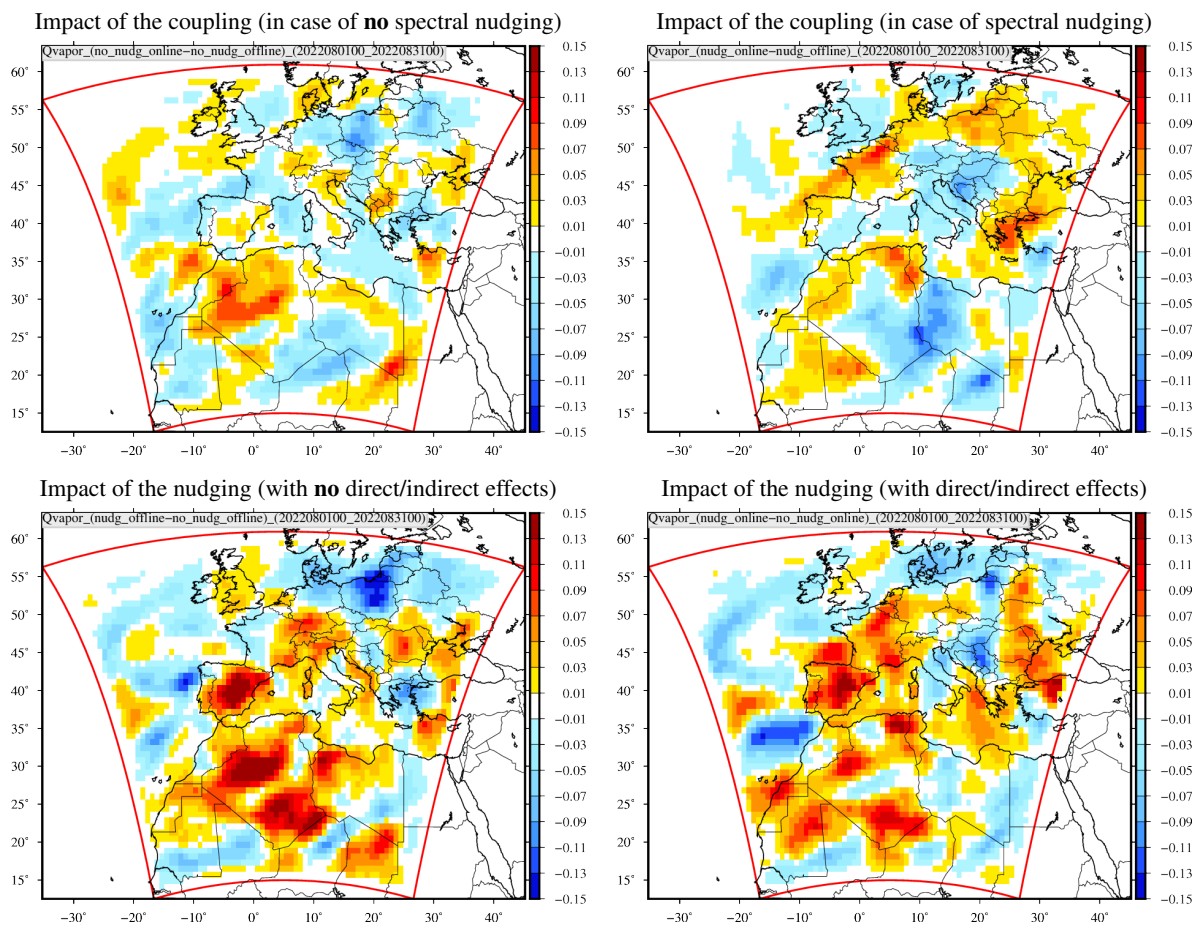

**Figure 7.** *Differences of vertically averaged water vapor mixing ratio (g/kg), time averaged over the period $1^{st}$:$31^{st}$ August 2022.*

or over land, no specific patterns are visible. Depending on the location and averaged over a month, the differences due to nudging can reach $\pm\ 6\ \mu g.m^{-3}$ for surface ozone concentrations.

The previous results are summarized in Figure 9 as distributions of the values displayed in the previous maps. The comparison of all differences as distributions enables to see the spread of these differences over the domain. For all variables, the
peak of the distribution is for the differences (nudg_online-nudg_offline, the green curve) and (nonudg_online-nonudg_offline, the red curve). These curves correspond to the simulations of the variability due to the coupling. The peaks indicate that these model configurations are those with the smallest differences. In addition, one can see that the differences are smaller with the green curve (nudging) than the red curve (no nudging). It means that the nudging reduces the variability of the simulations when comparing with or without coupling. The two other types of differences expressed the sensitivity of the model results to
the nudging with (nudg_offline-nonudg_offline, blue curve) and (nudg_online-nonudg_online, orange curve). In this case, the peak representing the small differences is reduced and numerous large differences are calculated, both negative and positive.



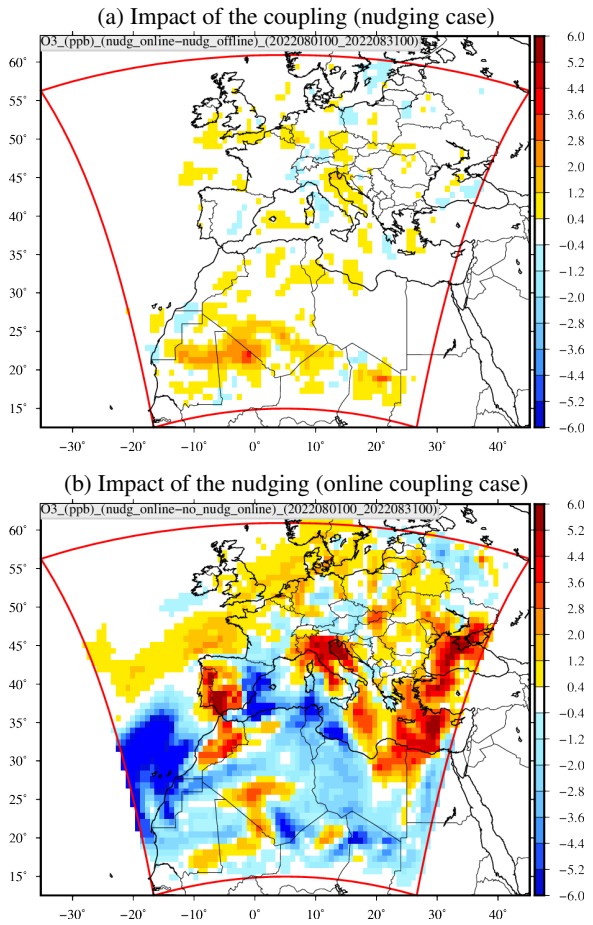

**Figure 8.** *Differences of ozone surface concentrations ($\mu g.m^{-3}$), time averaged over the period $1^{st}$:$31^{st}$ August 2022.*

It means that independently of the coupling, the nudging causes many more differences than the coupling. This is the case for meteorological variables and surface pollutants concentrations.

Additionally, results are also synthetized as mean averaged differences extracted from the maps of differences presented in the previous Figures. Results are in Table 4 and the goal is to try to extract an information about the variability of the coupling in case of nudging or not. The time-series and the distributions previously showed that the differences between the simulations offline and online are larger when there is no nudging than when there is nudging. The values in this Table are here to quantify it. The mean differences are first calculated using the sign values. But as the distributions showed that there is a large variability between negative and positive differences over the entire domain, we also add the mean differences calculated using absolute values of differences.

For each variable, it is interesting to compare the two lines: in each case, the difference is between offline and online simulation, and for the case of no nudging and the case of nudging. For all variable, the differences with no nudging are larger



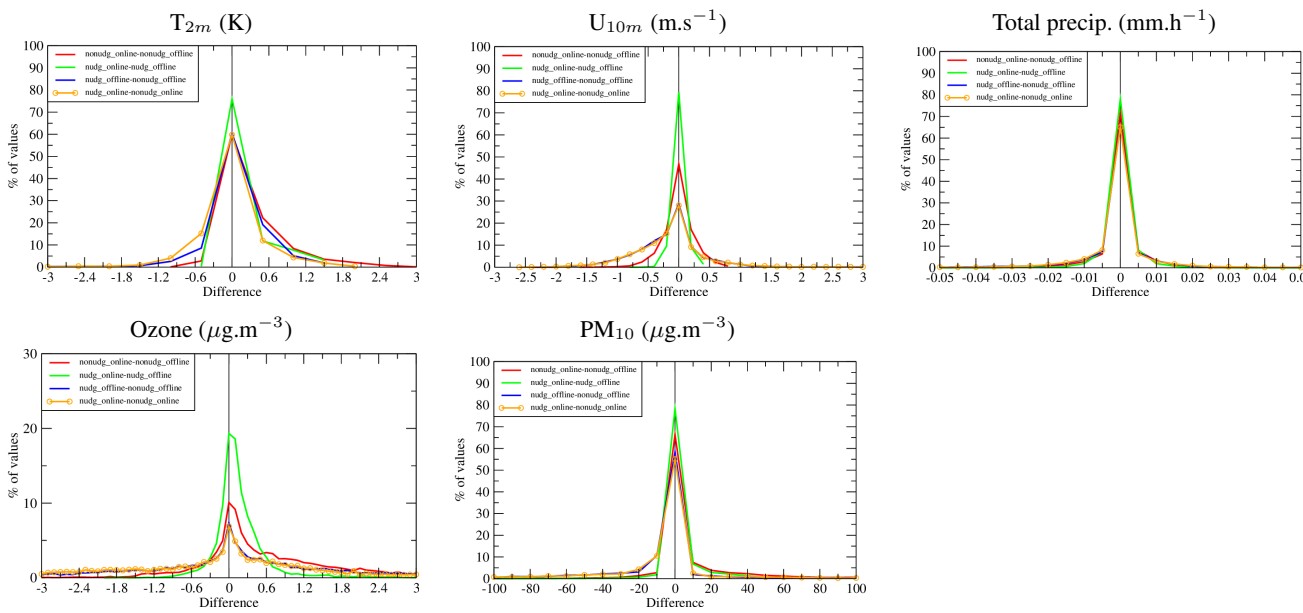

**Figure 9.** *Histogram of differences values, time averaged over the period $1^{st}$:$31^{st}$ August 2022.*

than the differences with nudging. It is observed both for the simple differences or the differences calculated with the absolute values. It means that the nudging reduces the variability of the simulations when they are online compared to those offline.

## 3.6 Vertical cross-sections

Another point of view for the meteorological variables is displayed in Figure 10 with temperature vertical cross-sections. Values are displayed along the latitude (from 15 to 55 $^o$N) and for the iso-longitude value of 5 $^o$E. Data are time-averaged between 10 August 00:00 UTC and 12 August 00:00 UTC. For the impact of the coupling, changes are more important in case of no spectral nudging and close to the surface and in altitude, mainly between 5000 and 8000 m. Changes are approximatively between -1 and +1 $^o$C. The most important changes are in altitude with maximum values around +1.5 $^o$C. For the impact of the nudging, changes are much more important but are similar with or without the online effects. The changes are not located at the same place that for the impact of the coupling: negative values are found of $\approx$ -1.8 $^o$C when they were positive in the other case. Large positive values are in the boundary layer and in the free troposphere.

Vertical cross-sections at a constant longitude of ozone concentrations are displayed in Figure 11 (same place and period than in the previous Figure). As for the temperature, changes are more important in case of the impact of the nudging than in case of the impact of the coupling. Changes are mostly above the boundary layer and both negative and positive with an amplitude of $\pm$ 20 $\mu$g.m$^{-3}$. The vertical structures are different than for temperature, not showing a direct link between the two variables. The lowest changes are in case of coupling and spectral nudging, the most realistic configuration, when ozone varies less than $\pm$ 10 $\mu$g.m$^{-3}$ in the whole modelled atmospheric column, with very low values close to the surface.





| Simulation | mean bias | mean abs(bias) |
|---|---|---|
| O$_3$ | | |
| nonudg(on-off) | 0.614 | 0.859 |
| nudg(on-off) | 0.172 | 0.286 |
| PM$_{10}$ | | |
| nonudg(on-off) | 18.056 | 22.079 |
| nudg(on-off) | 8.530 | 10.214 |
| T$_{2m}$ | | |
| nonudg(on-off) | 0.299 | 0.356 |
| nudg(on-off) | 0.195 | 0.239 |
| U$_{10m}$ | | |
| nonudg(on-off) | -0.005 | 0.170 |
| nudg(on-off) | 0.003 | 0.064 |
| Precipitation | | |
| nonudg(on-off) | -0.468 | 4.161 |
| nudg(on-off) | 0.101 | 1.815 |

**Table 4.** *Mean differences calculated with the values over the whole domain and corresponding to the mean averaged differences during the entire month of August 2022. Mean differences are calculated with the signed values and with the absolute values in order to avoid the effect of the negative/positive values, possibly reducing the mean average.*

The vertical cross-sections are also presented for PM$_{10}$ concentrations in Figure 12. Differences are less important between the four configurations than for the previous studied variables. The largest differences are still for the impact of the nudging. Absolute differences are limited to the boundary layer, with maximum difference values of $\pm$ 300 $\mu$g.m$^{-3}$. The location shows these differences are mainly for latitude lower than 30 $^o$N, indicating desert areas, then differences driven by mineral dust concentrations. In altitude, between 1000 and 3000 m, and above the Mediterranean sea (latitude 40$^o$N), negative values are
found, showing that the nudging reduces the concentrations. It is collocated with an increase of ozone which can be explained by the fact that less aerosols means more radiative fluxes and therefore more photochemistry.

## 4   Online coupling: impact of the spectral nudging

It has already been shown that the impact of nudging is far greater than that of online or offline coupling. In the following, we will therefore only present results for the online configuration, corresponding to the most realistic processes. The differences
will be calculated and presented only for the fact to use the spectral nudging or not.

     Results are presented in Figure 13 for the 2m-temperature (K), 10m wind speed (m.s$^{-1}$), mineral dust emissions (g.m$^{-2}$.h$^{-1}$) and AOD (no dim.) between spectral nudging or not, in online coupling mode, and time averaged over the period 1$^{st}$:31$^{st}$



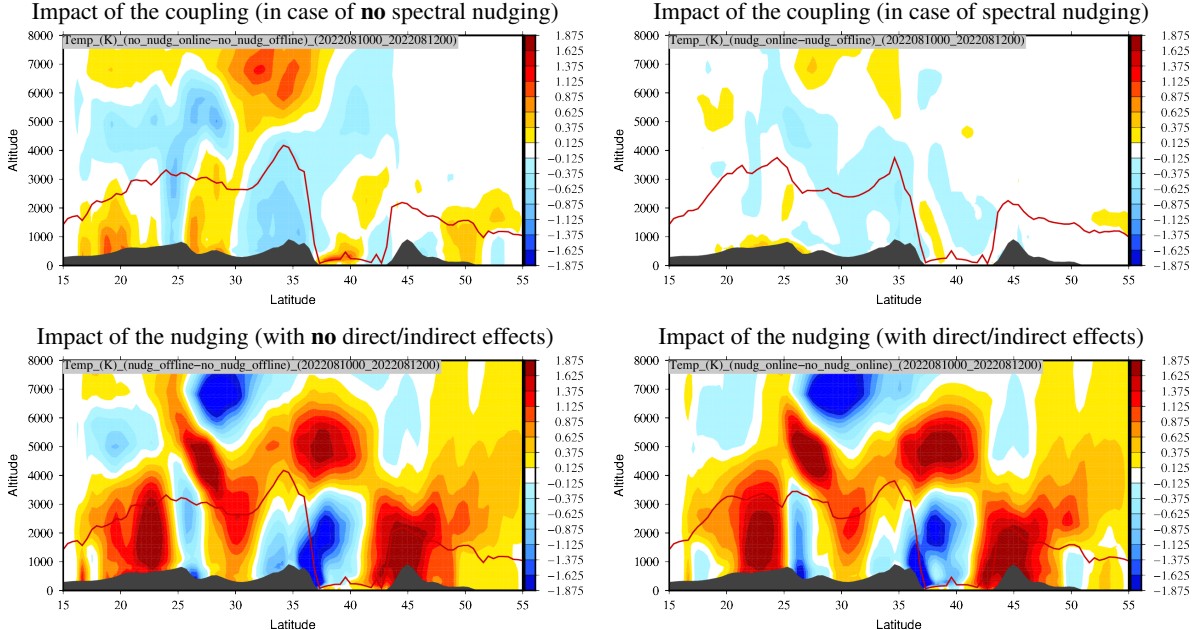

**Figure 10.** *Vertical cross-section of differences of temperature ($^\circ C$) between spectral nudging or not, coupling or not, and time averaged over the period $10^{st}$:$12^{st}$ August 2022. The red line represents the boundary layer height (m) of the simulation (a) if the difference is (a)-(b).*

August 2022. As for the previous presented variables, there is no systematic spatial patterns or coherent structures. This effect is always due to the fact that the results are presented as an average over a month, incorporating local changes and their

transport. But the important point is to assess their magnitude in terms of differences.

For temperature, the differences are both negative and positive and can reach $\pm$ 1.5 K. For the 10m wind speed, these differences are mainly negative (a reduction in wind) except over the sea, where local positive maxima can reach 2 m.s$^{-1}$. For mineral dust emissions, the differences are localised where these emissions occur, i.e. mainly in North Africa. The main trend is negative differences showing that, on average, nudging tends to reduce these emissions. The differences in AOD represent

a synthesis of the previous differences, this variable representing the aerosol load in the atmosphere and therefore reflecting changes in temperature and wind speed, and therefore dust emissions, their concentrations and therefore their optical thickness. There are wide spatial variations in AOD, with large positive structures over Africa, but also large negative structures over the south-western part of the domain, including a maritime area. The differences are important and around $\pm$ 0.15.

# 5 Conclusions

In this study, we have investigated the impact of the spectral nudging and coupling (aerosol cloud radiation) on regional simulations of atmospheric pollutants. These two processes are able to modify the meteorology, but not necessarily in the same

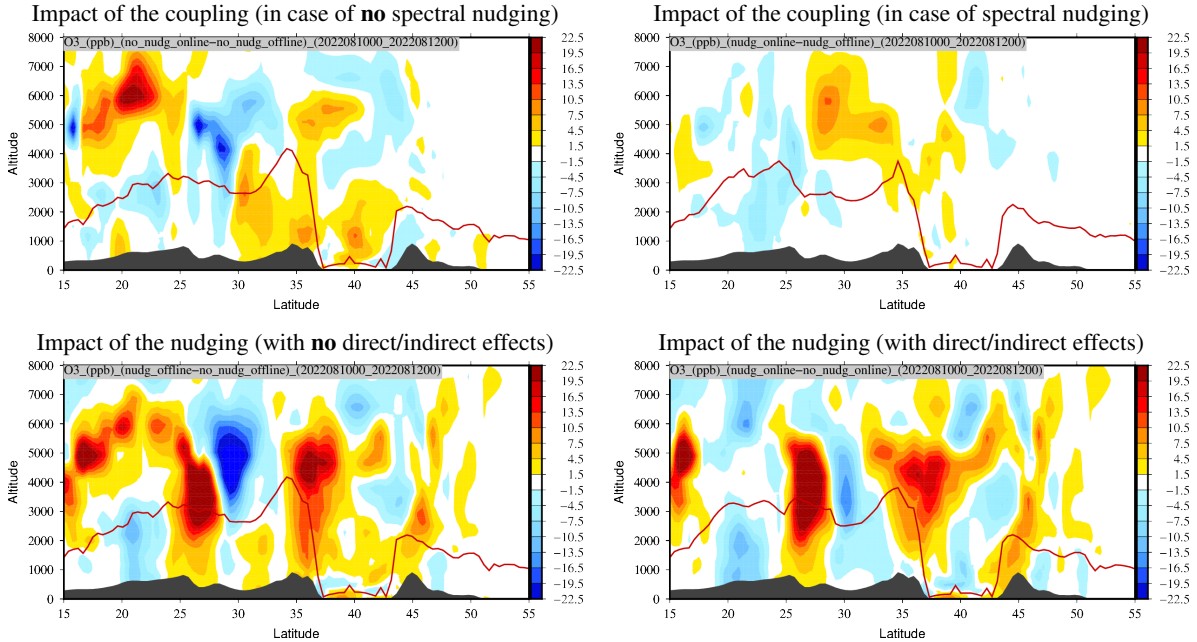

**Figure 11.** *Vertical cross-section of differences of $O_3$ ($\mu g.m^{-3}$) between spectral nudging or not, coupling or not, and time averaged over the period $10^{st}:12^{st}$ August 2022. The red line represents the boundary layer height (m) of the simulation (a) if the difference is (a)-(b).*

way. Their effects can be double-counted or contradictory, however, in both cases, they should better represent the reality we try to simulate.

To quantify this impact, we carried out four simulations, each lasting two months (during the summer of 2022) and covering

Europe and part of Africa. These four simulations were combinations with and without nudging and with and without coupling. The results show first of all that the four simulations differ from one another. For the pollutants studied, $O_3$, $PM_{10}$ and AOD, and in comparison with measurements, the simulations with nudging gives the best results, showing that, as expected, applying nudging permits to have simulation outputs closer to the observed data. At this point, the conclusions of the present study align with what is already known for climate models and extends it to chemistry-transport modelling.

We have also observed that the use of nudging reduces the sensitivity of the outputs on model configuration (in our case, the application of online coupling). As a consequence, the effect of coupling on the meteorological variables is smaller when nudging is applied, which was expected from previous studies (Pohl and Crétat, 2013), but also on the concentrations of gas-phase and particulate species (Table 4). In our case, the sensitivity of the model outputs to coupling is reduced by a factor ranging from 30 to 70% depending on the variables. While this might suggest that nudging could conduct to an underestimation

of the model sensitivity to coupling, it has been shown in climate modelling that applying nudging also gives more significance to the simulated sensitivity by dampening the internal variability of the meteorological model.



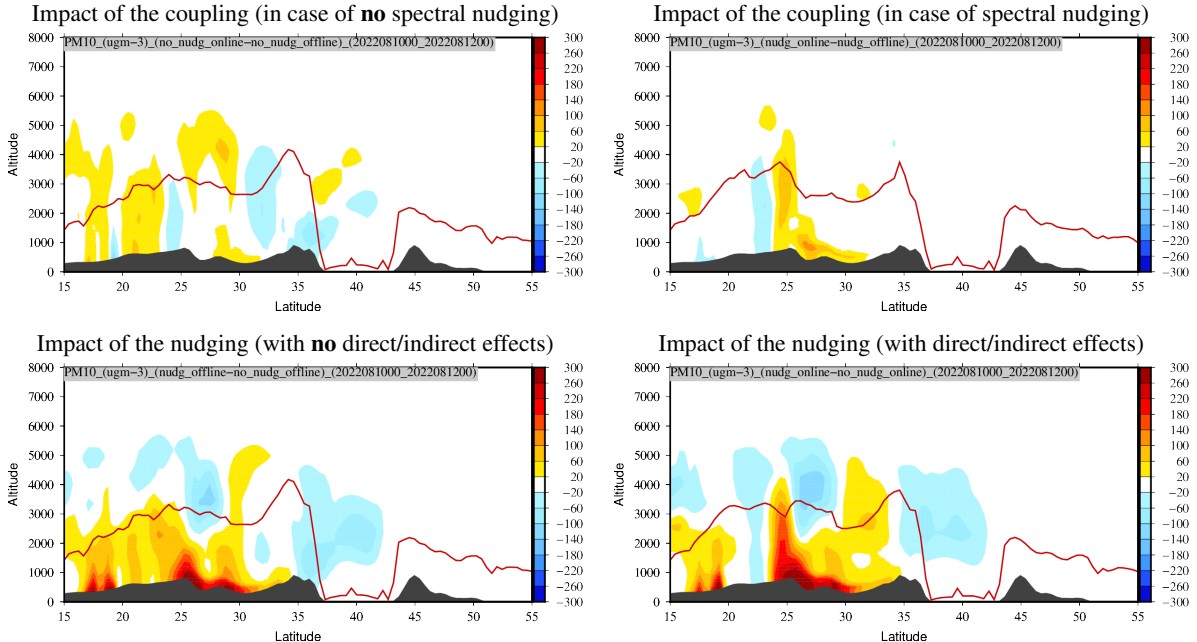

**Figure 12.** *Vertical cross-section of differences of $PM_{10}$ ($\mu g.m^{-3}$) between spectral nudging or not, coupling or not, and time averaged over the period $10^{st}$:$12^{st}$ August 2022. The red line represents the boundary layer height (m) of the simulation (a) if the difference is (a)-(b).*

The results of our study, summarized in Table 4 can be interpreted as ranges of sensitivity of the key variables in meteorology and chemistry-transport models to aerosol-meteorology feedback, with the sensitivity in the presence (resp. absence) of nudging giving an lower (resp. upper) boundary for the sensitivity of each variable to aerosol-meteorology feedback. The sensitivity

determined in the absence of nudging includes the effect of the feedbacks themselves but also of the internal variability of the meteorological model, while the sensitivity in the presence of nudging includes essentially the effect of the feedbacks, but possibly dampened by nudging. For example, in the present study, the sensitivity of $PM_{10}$ concentration to these feedbacks ranges between 10 and 22 $\mu g.m^{-3}$, that of ozone between 0.29 and 0.862 $\mu g.m^{-3}$ , and between 0.24 and 0.36 K for 2m-temperature. This conclusion is of course limited to the models used, the simulation domain and the period studied, as well as the parameters

chosen, particularly the nudging constants.

An important outcome of this study lies on the fact that the use of nudging and coupling options can have counter intuitive impacts when CTM are used to analyse the impact of emission reduction scenarios. For instance, it is very important to keep in mind that in the case of a coupled system, concentrations change not only because emissions change but also because meteorology is also affected. As a perspective of this study, simulations should be repeated with nested domains to address one

other dimension of the problem: the impact of the horizontal resolution.



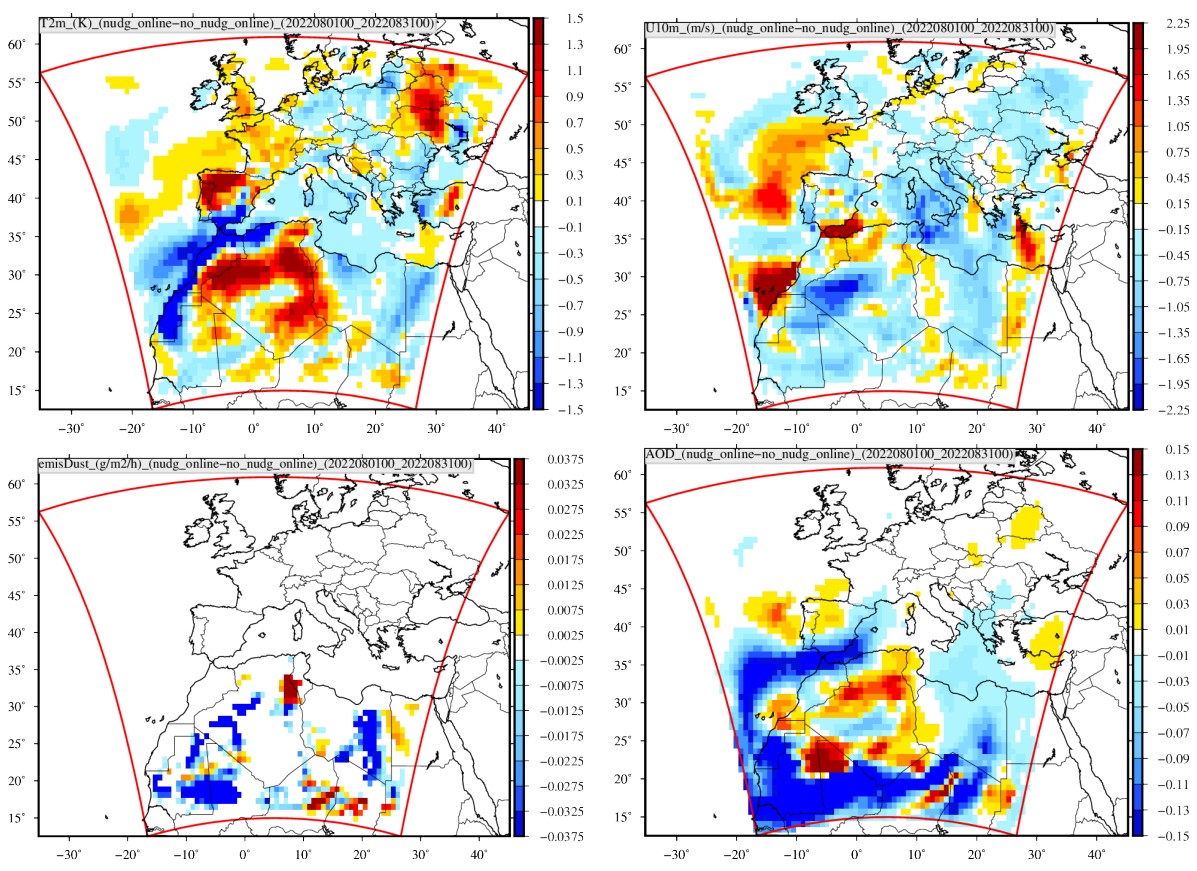

**Figure 13.** *Differences of 2m-temperature (K), 10m wind speed (m.s$^{-1}$), mineral dust emissions (g.m$^{-2}$.h$^{-1}$) and AOD (no dim.) between spectral nudging or not, in online coupling mode, and time averaged over the period from $1^{st}$ to $31^{st}$ August 2022.*

*Code availability.* The CHIMERE v2020 model is available on its dedicated web site https://www.lmd.polytechnique.fr and for download at https://doi.org/10.14768/8afd9058-909c-4827-94b8-69f05f7bb46d.

*Data availability.* All data used in this study, as well as the data required to run the simulations, are available on the CHIMERE web site download page https://doi.org/10.14768/8afd9058-909c-4827-94b8-69f05f7bb46d.

*Author contributions.* All authors contributed to the model development. LM coordinated the manuscript and all authors wrote a part and reviewed it.



*Competing interests.* The authors declare that they have no conflict of interest.

*Acknowledgements.* The authors the OASIS modeling team for their support with the OASIS coupler, the WRF developers team for the free use of their model. We thank the investigators and staff who maintain and provide the AERONET data (https://aeronet.gsfc.nasa.gov/).
European Environmental Agency (EEA) is acknowledged for their air quality station data that is provided and freely downloadable (https://www.eea.europa.eu/data-and-maps/data/aqereporting-8).



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

## Appendix A: List of measurements stations

| Site | Longitude (°) | Latitude (°) | Site | Longitude (°) | Latitude (°) | Site | Longitude (°) | Latitude (°) |
|---|---|---|---|---|---|---|---|---|
| **Meteorological stations** | | | Palma | 2.62 | 39.55 | Moerkerke | 3.36 | 51.25 |
| Madrid | -3.55 | 40.45 | Paris | 2.33 | 48.86 | Neuglobsow | 13.03 | 53.16 |
| Mallorca | 2.73 | 39.55 | Saada | -8.15 | 31.62 | OHP | 5.71 | 43.93 |
| Bordeaux | -0.7 | 44.83 | Saclay | 2.16 | 48.73 | OSavinao | -7.69 | 43.23 |
| Firenze | 11.2 | 43.8 | Toulouse | 1.37 | 43.57 | Payerne | 6.94 | 46.81 |
| Orleans | 1.75 | 47.98 | Vienna | 16.33 | 48.23 | Peyrusse | 0.17 | 43.62 |
| Lille | 3.1 | 50.57 | **Pollutants stations** | | | Peristerona | 33.05 | 35.03 |
| Salzburg | 13.00 | 47.80 | Aytre | -1.11 | 46.13 | PuertoCotos | -3.96 | 40.82 |
| Munster | 7.70 | 52.13 | Airvault | -0.13 | 46.82 | Preila | 21.06 | 55.35 |
| Stansted | 0.23 | 51.88 | Barcarrota | -6.92 | 38.47 | Rageade | 3.27 | 45.10 |
| Melilla | -2.95 | 35.28 | Biarritz | -1.55 | 43.47 | Rambouillet | 1.83 | 48.63 |
| Perugia | 12.50 | 43.08 | Burgas | 27.38 | 42.46 | Riom | 3.12 | 45.89 |
| Chievres | 3.83 | 50.57 | Brotonne | 0.75 | 49.49 | Starina | 22.26 | 49.05 |
| Bourget | 2.45 | 48.97 | Breazu | 27.54 | 47.19 | StDenisAnjou | -0.44 | 47.78 |
| Friedrichshafen | 9.52 | 47.67 | Carling | 6.76 | 43.43 | StMalo | -2.00 | 48.65 |
| **AOD stations** | | | Campisabalos | -3.14 | 41.28 | Solling | 9.55 | 51.70 |
| Arcachon | -1.16 | 44.66 | Diga | 7.24 | 45.43 | Schauinsland | 7.90 | 47.91 |
| Palaiseau | 2.20 | 48.70 | ElsTorms | 0.71 | 41.40 | Tremblay | 2.57 | 48.95 |
| Aubiere | 3.11 | 45.76 | Fontainebleau | 2.64 | 48.35 | Ulborg | 8.43 | 56.28 |
| Barcelona | 2.11 | 41.38 | Focsani | 27.21 | 45.69 | Uto | 21.37 | 59.77 |
| Birkenes | 8.25 | 58.38 | Germany | 8.90 | 48.64 | Valentia | -10.24 | 51.93 |
| Coruna | -8.42 | 43.36 | Hunsr | 7.19 | 49.74 | Viznar | -3.53 | 37.23 |
| Evora | -7.91 | 38.56 | Illmitz | 16.76 | 47.76 | Verneuil | 2.61 | 46.81 |
| Kanzelhohe | 13.90 | 46.67 | Iskrba | 14.86 | 45.56 | Valderas | -5.44 | 42.07 |
| Lampedusa | 12.63 | 35.51 | Kergoff | -2.94 | 48.26 | Vredepeel | 5.85 | 51.54 |
| Lille | 3.14 | 50.61 | LaTardiere | -0.74 | 46.65 | Vosges | 7.12 | 48.49 |
| Loftus | -0.86 | 54.56 | Kosetice | 15.08 | 49.58 | Vredepeel | 5.85 | 51.54 |
| Madrid | -3.72 | 40.45 | LaTardiere | -0.74 | 46.65 | Vysokoe | 23.43 | 52.33 |
| Murcia | -1.17 | 38.00 | LeCasset | 6.46 | 45.00 | Waldhof | 10.75 | 52.80 |
| Messina | 15.56 | 38.19 | Lahemaa | 25.90 | 59.50 | Zoodyss | -0.39 | 46.14 |
| Murcia | -1.17 | 38.00 | Mera | -0.45 | 48.64 | Zorita | -0.16 | 40.73 |
| Napoli | 14.30 | 40.83 | MontsecOAM | 0.72 | 42.05 | | | |

**Table A1.** *Characteristics of measurements stations used in this study.*