# Peer review of "How is the relative impact of nudging and online coupling on meteorological variables, pollutant concentrations and aerosol optical properties?"

_Geoscientific Model Development, 2023_

## Author Response (AR1)

Article gmd-2023-209:
**How the meteorological spectral nudging impacts on aerosol radiation clouds interactions?**
L.Menut et al.

**1   Answer to the Editor**

Dear Editor and Reviewers,

Thanks for your interesting comments. We have responded to all the comments made by the two reviewers. Most of the requests resulted in changes in the manuscript. For reviewer #1, clearly unfamiliar with regional modeling, meteorology-chemistry couplings and nudging, we have done the maximum to add details, but not everything can be completely redefined, and references are here for this. In particular, a request to add a reference seemed out of context to us, and the reviewer didn't explain the point of doing so. For reviewer #2, with more questions of substance than of form, improvements were made in the presentation of results. Figures and analysis of the geophysical processes have been added explaining the maps and scores.

Best regards,
Laurent Menut
March 11, 2024

**2   Answers to the Reviewer 1 (9 Jan 2024)**

This paper presents an analysis on the impact of meteorological nudging and online coupling of aerosol microphysics on selected simulated variables in a regional model.
The results are not very surprising, but still useful for model applications, e.g., for assessment studies. In that sense, I find this study within the scope of GMD.
There are, however, several issues that should be addressed before the paper can be accepted for publication:

We would like to thank the Reviewer for his precise proofreading and the many comments. All comments have been taken into account.

1. The Results section (Sect. 3) is rather descriptive and no attempt is done by the authors to interpret the results. Some specific features noted on the plots need to be explained in view of the model parametrizations and configurations. This is done only for a few specific cases, but it is missing otherwise.

   The section 3 starts with "statistical scores" to quantify the impact of the two studied processes. The differences are discussed. The next sub-sections, 3.2 and 3.3 are about time-series and all plots are analyzed. Perhaps, for the reviewer, the problem is that the discussion is not about the absolute differences. For this specific study, it is normal and due to the fact that two different way of modelling are compared: one can not directly compare their physical impact, because, here, this is two different modelling approaches. Thus, the discussion is in term of realism of the results and model variability. Results are discussed in this way, and, we agree, it is not classical compared to other studies when a sensitivity study on emissions or temperature will discuss the physical impact on concentrations, knowing precisely the role of emissions and meteorology on

the chemistry. The same remark applies for the maps presented in section 3.5: here, having a mean time-averaged map, it is not realistic to discuss about processes and in detail on the physico-chemical causes and consequences. It is also why, at the end of this section, results are presented as distributions of differences: the goal of the study is mainly to quantify the relative impact of two different modelling approaches.

2. Although the authors have access to a quite comprehensive set of stations data (Table A1) they choose to show and discuss only a few stations and their choice is not very well motivated. I think it would be valuable to consider the data from all available stations and try to derive a synthesis from them, for example by applying some kind of statistical metrics to summarize the results and trying to derive a more general conclusion.

The main goal of an article is to be synthetic and conclusive. The data from all available are taken into account, in the statistical scores calculations. The Figures are chosen as representative examples of the obtained results. There is no interest to publish a catalog when the examples are here to illustrate already the results. However, for the selected stations, we added some explanations about the choice, for example, by describing their position in relation to the fires studied and the plumes observed. Note also that the test case is the same than in our previous publication where all stations were described and where the events were already discussed. The following text was added in the "Measurements data: section:

> The results will be presented in two different formats: general statistics to show the trend of impacts on simulated values, and examples of time-series and maps to illustrate these statistics more precisely. The measurements stations chosen as examples are selected for their representativeness in relation to the other stations, as well as for their geographical position in relation to the processes studied.

and more precisely the following sentences were added in each subsection where measurements stations are used:

> Here, we present examples for two stations in France, Orléans and Bordeaux, located close to the studied fires. Bordeaux is the closest station to the studied Landes fires and directly under the fire plume and Orléans is located at 400 km to the north-east of Bordeaux, but also under the fire plume.

For Biarritz and Fontainebleau, the desciption is already in the present manuscript as well in the cited article, (Menut et al., 2023).

3. The terminology used to distinguish the coupled / uncoupled model configuration is not consistent, sometimes different terms are used (online/offline, direct/indirect effects). The authors should aim at a more consistent definition of the tested configurations.

Ok, the manuscript was revised to have a more clear terminology. Note that online/offline and coupled/uncoupled is the same. The word "coupled" was used three times in the manuscript and it was changed. But, direct and indirect are both in the 'coupled' category.

4. I find the title misleading: the study is not addressing the impact of nudging on aerosol-cloud-radiation interactions, but rather the impact of model coupling on meteorological variables, pollutants concentrations and aerosol optical properties. Please consider a more precise title.

Yes, we agree. And more precisely, it is about "the relative impact of nudging and coupling on meteorological variables, pollutants concentrations and aerosol optical properties". The title was

changed and is now "How is the relative impact of nudging and online coupling on meteorological variables, pollutants concentrations and aerosol optical properties?"

5. The presentation quality should be improved: several sentences are unclear and/or hard to read and some of the figures have small fonts or unconvenient colors (see detailed suggestions below).

   The manuscript was carefully read and many corrections were done in this revised version. The colors are adapted to the results and are classical: for the differences maps, widely used, in this study, it is very important to quickly see the negative versus positive values. The zero difference is white, negative differences are with cold colors and positive differences with warm colors. It is used in many publications.

**Specific comments**

- L5: WRF-CHIMERE coupled model: I would rather write WRF-CHIMERE regional model, in its coupled and uncoupled configuration.

  OK it was corrected.

- L13: I would explain why, or for which use case, a forcing inside the domain is needed.

  It is the definition of the nudging. Perhaps the sentence is not clear and it was was reworded. The nudging is not mandatory. Some models with high resolution are not using it. But in general, it is not really explained in publications, then difficult to cite here.

  > Being regional, the two models need forcing at the boundaries of the domain and inside the domain. For the meteorological part and inside the domain, the technique used is called nudging and it could be 'grid' or 'spectral', (von Storch and Zwiers, 2001), (Kruse et al., 2022).

- L18-19: it would be good to explain how aerosol can modify the meteorology in the model.

  It is the basis of the direct and indirect effects modelling. Some sentences were added. A simple explanation was added in the CHIMERE model description section.

  > This v2020r3 version of CHIMERE is to date the last distributed one and is designed to take into account the direct and indirect effects of aerosols on cloud and radiation (the online mode) or not (the offline mode). The way these effects are taken into account is described in Briant et al. (2017) (for the direct effects) and Tuccella et al. (2019) (for the indirect effects). Mainly, the direct effect corresponds to the attenuation of radiation by aerosol layers, and the indirect effect corresponds to cloud formation by the presence of fine particles.

- L20: it is not clear which forcing is meant here.

  The forcing is the global forcing and 'fine' was replaced by 'large' (the sentence was not correct). It was corrected in the text.

  > These changes are performed at higher spatial and temporal scales than the global forcing which is intrinsically a large scale process.

- Fig. 1: the caption needs to provide more details to help the reader understanding the figure. For example, what is the meaning of the different colors and dashes of the arrows?

  Yes, we agree. The caption was extended and is now self-consistent. But many explanatons were already in the text for this Figure. There is now two ways to have the explanations: caption and text. It is now:

The paradox of the regional model nudged by a global model. The global model performs a meteorological simulation, generally including aerosol climatology to take into account the direct and indirect effects of aerosols. If the simulation is a reanalysis, there may also be data assimilation, such as optical thickness estimated from satellite observations. But the overall simulation will have included aerosols in the meteorological calculation. This global simulation will serve as a forcing for the regional simulation. The regional meteorological model will serve as a forcing tool, or will be coupled to the chemistry-transport model calculating aerosols. Aerosols are also taken into account, but at a different resolution. The black grid is the global model and the blue grid the regional model. The dotted arrow indicates that aerosols may not be the same species or have the same size distribution, depending on the chemistry-transport and climatology models used.

- L30: this methodological alternative should be discussed again in the conclusions in view of the analysis presented in the paper. Since the paper is mainly addressed to the users of this model, a key-message should be formulated to help them choosing the propert methodology in the future.

  Yes, we can discuss again about this paradox. But, this paper is not mainly adressed to the users of this model. It is a general problem of all regional models, using boundary conditions, then nudging and nesting. It is true for atmospheric, troposphere and stratosphere, or ocean models. And finally, the choice is left up to the user, and mainly to the case study he wants to carry out. Depending on the horizontal resolution and spatio-temporal variability of the process under study, the choice will be different. The only real constraint is comparison with observations.

- L50: in this paragraph, you may consider mentioning Chrysanthou et al. (2019) too (https://doi.org/10.5194/acp-19-11559-2019).

  This study was on a close topic but for the stratosphere and for climate modeling. It is why it was out of the scope for our bibliography. Note sure it is really suitable and relevant for this study. Why the reviewer considers this reference has to be addded in this study? When a reviewer asks for an addition of a reference, it could be useful to justify it (to avoid self promotion and be sure that the reviewer is not one of the co-authors).

  Chrysanthou, A., Maycock, A. C., Chipperfield, M. P., Dhomse, S., Garny, H., Kinnison, D., Akiyoshi, H., Deushi, M., Garcia, R. R., Jöckel, P., Kirner, O., Pitari, G., Plummer, D. A., Revell, L., Rozanov, E., Stenke, A., Tanaka, T. Y., Visioni, D., and Yamashita, Y.: The effect of atmospheric nudging on the stratospheric residual circulation in chemistry-climate models, Atmos. Chem. Phys., 19, 11559-11586, https://doi.org/10.5194/acp-19-11559-2019, 2019.

- Fig. 2: the two letters identifying the stations are hard to read. It might be wiser numbering the stations instead (here and in Table A1). The choice of the colors is not optimal for color-blinded readers. Please consider alternative colors.

  The Figure have been reprocessed. But interest of PDF files (such as with GMD) is the capability to zoom on a Figure. Indeed, it is really difficult to represent stations locations and names on a single map. We think the colored symbols and the two first letters are the best way to visualize where are the stations of interest and to remember their location when you know their name (a number doesn't help). To be more easy to read, we changed the symbols and now meteorological stations are represented with blue squares, Aeronet stations with red diamonds and pollution stations with green circles. The symbols are smaller and the letters are larger. The caption was changed also.

- L84: I would suggest naming Sect. 2.1 Model parameterizations and providing a few more details about the parametrizations themselves. For example, about the cloud scheme and how this can be coupled to the aerosol microphysics (which I guess is part of CHIMERE described below).

  It is not only 'parameterizations' but the model set-up. The title seems correct here. About more details, they are already published in other previous articles. For example, for the coupling, it is

described and cited in the next paragraph about the CHIMERE model.

- L96: I am not sure that grid nudging applied over all grid cells is necessarily an advantage, since this does not allow to limit the nudging to certain scales. Moreover, the fact that spectral nudging is less intrusive depends on its configuration and whether, for instance, wave-0 nudging is considered or not. Please clarify.

  Yes, right. "advantage" is replaced here by "characteristic"

- L103: what do you mean with perturbation of the geopotential? What is the source of this perturbation? Please specify.

  Yes, in this context, it is just 'geopotential height' without perturbation.

- L113: I find the remaining of this section quite hard to follow. I would suggest providing clear mathematical definitions for the nudging coefficients and to be more specific on how the calculations of the wave numbers (Eq. 1) are considered in the nudging scheme. Is this a maximum wave number for which spectral nudging is applied?

  The definition of the nudging is completely exposed and discussed in the dedicated publications, already referenced in this article. The Reviewer is invited to see the bibliography, about this point, it is clearly not the goal of this article to repeat these mathematical concepts.

- L133: I would add an introductory sentence to make clear the CHIMERE is an aerosol-chemistry scheme which is used when running WRF in online mode (if I understood correctly). Is this online mode only considering the impact of aerosol or also the impact of chemistry (e.g. ozone impact on radiation)?

  Unfortunately, it is not that. CHIMERE is not a scheme but a chemistry-transport model. L.77: "The two models used...". But, we agree that the coupling is probably not enough described and we added a paragraph about this point. The CHIMERE model is always running, in online or offline mode since the goal of this study is to model pollutants.

- L133: to take into account the direct effects of aerosol on clouds and radiation. This sentence is unclear, also in relation with the following sentence. If I interpreted it correctly, you mean that CHIMERE can account for both aerosol-radiation and aerosol-cloud interactions. If this is the case, I would also add a short explanation on how this is achieved technically. Referring to other publications is not enough, the text should be self explanatory, at least concerning the basic functionalities of the model.

  Yes, a paragraph was added about this point. It seems that the references are not enough here. IN addition, to the previsouly added lines in the CHIMERE description, the following is added:

  When used with the meteorological WRF model, CHIMERE and WRF are coupled using the OASIS-MCT coupler. As in offline mode, WRF send hourly meteorological fields for chemistry-transport to CHIMERE and CHIMERE send aerosols and Aerosol Optical Depth fields to WRF for the radiation attenuation and the microphysics.

- L140: the aerosol microphysics simulated by the model should also be mentioned (which processes and species are considered?).

  There is no microphysics calculations in CHIMERE. It is now explained in the new paragraph.

- L158: please list the applied statistical scores here with their full name (the table just show their symbol).

  Yes, OK, a paragraph was added also to define the statistical scores.

They are defined as follows. The variables $O_t$ and $M_t$ stand for the observed and modeled values, respectively, at time $t$. The mean value $\overline{X_N}$ is defined as:

$$\overline{X_N} = \frac{1}{N} \sum_{t=1}^{N} X_t \tag{1}$$

with $N$ the total number of hours of the simulation. To quantify the temporal variability, the Pearson product moment correlation coefficient $R$ is calculated as:

$$R = \frac{\frac{1}{N}\sum_{t=1}^{N}(M_t - \overline{M_t}) \times (O_t - \overline{O_t})}{\sqrt{\frac{1}{N}\sum_{t=1}^{N}(M_t - \overline{M_t})^2 \times \frac{1}{N}\sum_{t=1}^{N}(O_t - \overline{O_t})^2}}, \tag{2}$$

The spatial correlation, noted $R_s$, uses the same formula type except it is calculated from the temporal mean averaged values of observations and model for each location where observations are available.

$$R_s = \frac{\sum_{i=1}^{I}(\overline{M_i} - \overline{\overline{M}})\,(\overline{O_i} - \overline{\overline{O}})}{\sqrt{\sum_{i=1}^{I}(\overline{M_i} - \overline{\overline{M}})^2\,\sum_{i=1}^{I}(\overline{O_i} - \overline{\overline{O}})^2}} \tag{3}$$

where $I$ is the number of stations. The Root Mean Square Error (RMSE), is expressed as:

$$\text{RMSE} = \sqrt{\frac{1}{T}\sum_{t=1}^{T}(O_{t,i} - M_{t,i})^2} \tag{4}$$

To quantify the mean differences between the several leads, the bias is also quantified as:

$$\text{bias} = \frac{1}{N}\sum_{t=1}^{N}(M_t - O_t) \tag{5}$$

- L162: please provide an example of such local features (orography?).

  Yes, right, for example the orography. It was added in the text.

  It is not the case for the wind speed, a more "local" variable more influenced by local features such as orography, vegetation type and height.

- L165: the last sentence of this paragraph is unclear. Please consider rephrasing.

  Yes, OK. The modified sentence is: "The differences in scores between online and offline are smaller, and the offline configuration gives the best results."

- Table 2: please be more specific on the procedure applied to aggregate the statistical scores.

  Perhaps the word "aggregated" is not the best choice. The caption was modified as "Scores are calculated..." and the calculation is now explained with the new paragraph of definitions and equations. The new caption is:

  Scores are calculated for all stations and over the entire modelled period (July and August 2022). The best scores values are framed.

- L174: The impact of the coupling is less important and the scores are more or less the less with and without the coupling. Do you mean, the scores are quite similar independently of the coupling mode?

  yes, that's right.

- L179: ...there is no clear impact of the use of the direct effects or nor on the scores. This is an example where the use of different terminology (main comment 3 above) makes the text hard to understand. What do you mean with the use of the direct effects? Are you referring to coupled vs. uncoupled model? Please clarify.

  Here, it is necessary to add details. The impact of coupling on AOD is mainly driven by the direct effects. The indirect effects also act but in a lesser extent. A new sentence is proposed here: "As for surface concentrations, there is no clear impact of the use of the coupling (with direct and indirect effects) or not on the scores."

- L188: what is the reason for showing and discussing only the stations data for Orléans and Bordeaux? As noted in the main comment 2 above, you have a valuable set of measurements and you should take full advantage of them.

  We take advantage of all measurements: they are all considered in the calculation of the statistical scores. The time series are in addition and represent examples to better see the differents model configurations and the measurements variability. The stations were selected as examples of representative sites considering their distances to the fires and the advected plumes. It is, for example, exactly the same type of comparisons as in many publications, including publication of our teams such as:

  - Menut, L., Cholakian, A., Siour, G., Lapere, R., Pennel, R., Mailler, S., and Bessagnet, B.: Impact of Landes forest fires on air quality in France during the summer 2022, Atmos. Chem. Phys., 23, 7281-7296, 2023
  - Menut L., G.Siour, B.Bessagnet, A.Cholakian, R.Pennel and S.Mailler, Impact of wildfires on mineral dust emissions in Europe, Journal of Geophysical Research Atmospheres, 127, e2022JD037395. https://doi.org/10.1029/2022JD037395,

- L190: I would consider adding the results for the other stations to a supplement.

  The results are already included in all Tables of statistical scores. There is no need to add a catalog of time series, we will have no further scientific discussion relevant to the analysis of the results.

- Fig. 3: the left and right panels have the same titles, but show different variables. Please add the variable name to the title, e.g. Bordeaux (daily mean) - Temperature or similar. Do the right panels show the 10m wind speed or only its u-component (as indicated on the y-axis)? Please specify. The legend is very small and hard to read: since it is the same for all panels, you may consider a common legend at the bottom of the plot. Please also use more friendly colors (see above comment on Fig. 1). The same applies to Fig. 4.

  The same title because the title is the location. But the Figure have different y-axis with the name of the variable. This is the 10m wind speed as already specified in the text. L.201 for example "...for the 10m wind speed...". And for the colors, it is not easy to make different. We can add different symbols on each curve but it becomes quickly unreadable. And we think it is a norm to have symbols for measurements and lines for model outputs. For the labels, the Figure are processed independently, one by one, it is not possible to add one legend for all Figures. And for the readibility, it is easy to zoom with a PDF if it is difficult to read (that is not our case).

- L199: what is meant with interface here?

  Ok, we understand. The word "interface" is changed by "The model is able to follow this weather change except around the day of 15..."

- L203: why do the nudged simulations performed better, given that wind nudging is not effective at small scales? Or am I missing something?

  Yes, it is logical and means that local meteorology is also driven by large scale motions and not only by local turbulence or transport.

- L214: It is difficult to disentangle the several simulations and to diagnose the best scores without statistical calculations. Why not performing such analysis? These scores have been computed for temperature and wind speed, it should not be too difficult to repeat the analysis for ozone, PM and AOD.

  Yes, indeed: all statistical scores are already in the Table 3.

- L234: as above, please motivate the choice of these three stations. Although, I note again that including all stations would significantly improve the analysis.

  Some comments were added about the choice of the stations.

- L240: How are the direct/indirect effects impact the AOD? It is actually the other way around, the AOD determines the direct effect. Or do you mean that the coupled model version has an impact on AOD, due to the different representation of aerosols? This is unclear. Using consistent terminology, as noted above, would help understanding this.

  The impact of direct/indirect effects are extensively discussed in two references (already cited) in the case of the WRF and CHIMERE and in many other studies with other models. But some sentences were added in the model presentation section.

- L241: please try to elaborate on the possible reasons for these differences.

  It means that the model produces too much dust for these days (Birkenes is close to the desert where dust are emitted), then creating a bias in the Angstrom exponent. This bias is reduced with the nudg-online model configuration, probably because these two forcings are able to better represent the wind speed and direction. A sentence of explanation was added in this section.

- L256: same here, please include some insights to support the interpretation of the model results.

  Same reason than for the previous remark.

- L269: this paragraph is also quite descriptive and more interpretation would be useful.

  The following text was added:

  > Results are presented in Figure ?? for the water vapor mixing ratio, this variable is particularly important for the radiative transfer particularly at night. Water vapor as a radiative forcer contributes significantly to the greenhouse effect, between 35% and 65% for clear sky conditions and between 65% and 85% for a cloudy day as reported in (Bessagnet et al., 2020) and reference there in. The water vapor concentration fluctuates regionally and locally as shown particularly in the land/water transition bands and in mountainous areas. In these later regions, at night the long-wave radiation is one of the most important variable governing the radiative budget, a change of water mixing ratio initiated in the bottom of valleys by small motions immediately modifies the radiative balance.

- Fig. 9: please speficy that this histogram considers surface values.

  OK, done for ozone and $PM_{10}$. The temperature is at 2m heigt and wind speed at 10m height above ground level. The precipitation is the surface measurement.

- L368: these values are not very useful, without a reference to compare with. You may consider showing their relative counterparts as well.

  All models are different, all configurations are different, there is no "reference" for such variability. The values are useful that express that these model configurations have an important impact of the modelling of these pollutants or temperature.

- Table A1: the caption says that the characteristics of the stations are shown, but it actually shows only their location. If the information about the characteristics (e.g., urban, rural, etc.) is available, please add it.

The "characteristics" here are the name and location (longitude, latitude). The caption was changed accordingly by: "Name and location of measurements stations used in this study."

**Technical corrections**

All following corrections were taken into account. Thanks to the reviewer.

- L3: radiations → radiation.
- L51: the nudging → nudging.
- L68: at → on.
- L73: modelled pollutants surface concentrations → modelled surface concentration of the pollutants.
- L73 and following: The Section → Section.
- L97: when → while.
- L97: The measurements data → Measurement data.
- L209: mi-July → mid-July.
- L210: mi-August → mid-August.
- L272: may → may be.
- L351: strictly speaking, AOD is not a pollutant.
- L383: the authors → the authors thank.

**3 Answers to the Reviewer #2 (12 Feb 2024)**

**3.1 General Comments:**

This study compares the impacts of two forcing mechanisms for meteorological fields generated for regional air quality simulations, i.e. spectral nudging of modeled meteorology towards reanalysis fields and feedbacks from aerosols modeled by the air quality model on radiation calculations performed by the meteorological model when air quality and meteorology models are coupled. This topic is of interest to the regional air quality modeling community and very few earlier studies have attempted to address it. The general dominance of the nudging effect over the feedback effect is not unexpected but nevertheless valuable to document in a manuscript, though as suggested in one of my comments below some additional analysis could be performed to assess whether there are exceptions to this general conclusion. The modeling approach employed in this study is straightforward and sound. My main concern with the manuscript is that there is only limited motivation for many of the analysis choices made in Sections 3-4 (i.e. the selection of stations, time periods, latitudinal cross-sections, etc.) and insufficient interpretation of the results shown in the Figures and Tables in terms of the physical and chemical processes causing these results. I would also suggest including a comparison of the four simulations for time periods and locations with the largest aerosol coupling effects (as simulated by the no-nudging configuration). Stratifying results by hour-of-day may provide further mechanistic insights. Finally, the writing of the manuscript would benefit from careful proofreading for language and grammar to improve its clarity and readability.

We thank the reviewer for her/his analysis. He/she has clearly seen the essence of this study, which is to compare the effects of two very different numerical processes that are not comparable in nature. The detailed analysis of the impact of each process is well known in principle, whether it is nudging or direct and indirect effects. However, it is more difficult to carry out an analysis on a case study and a region, as the effects vary greatly in time and space. We have, however, added information on this point. A careful re-reading has also been carried out.

**3.2 Specific Comments**

- Line 2: "nudging": for readers not familiar with this term, it might be useful to expand this to "nudging of modeled meteorology towards reanalysis fields". It might also be worth considering to add "often" before "involve" because not all regional-scale model applications utilize either nudging (many applications use frequent meteorological restarts instead) or meteo-chemical coupling.

  Ok, the first sentence in the abstract was corrected.

- Figure 1: the interpretation of the arrow and associated text boxes ("resolution", "species", "size distrib") between the pink aerosol boxes is a bit unclear. Is it meant to imply a contradiction between the representation of these aspects in the regional scale vs. global scale context?

  The caption was rewritten to be more complete and more clear. It was also a comment from Reviewer #1. And yes, the dotted arrows are also there to show that in terms of aerosols from one resolution to another, and therefore possibly from one model to another, can lead to additional errors in the calculation.

- Line 152: Does "daily average" imply averaging over zero nighttime values? Or are nighttime values simply not reported and modeled?

  The expression 'daily average' is indeed delicate in the case of photometer measurements. The available measurements values are averaged over a 24-hour period, from midnight to midnight. Only the corresponding values are considered with the model. This sentence was added in the manuscript.

  > The AOD at a wavelength of $\lambda$=675 nm is daily averaged and compared to daily averaged modelled values. The available measurements values are averaged over a 24-hour period, from midnight to midnight. Only the corresponding values are considered with the model.

- Lines 153-154: Are no meteorological observations with higher precision reporting available for this analysis? This database is the concatenation of all available operational meteorological surface data. It is one the best database, also because data are precisely checked.

- Lines 161 - 165: These sentences are an example of where the writing of the manuscript could benefit from careful proofreading for language and grammar.

  OK. This paragraph was rewritten as:

  > For these two variables, 2m temperature ($T_{2m}$) and 10m wind speed ($u_{10m}$), the best scores are obtained for the simulations with spectral nudging, but not systematically for the simulation with the coupling. The scores reflect the spatial and temporal representativeness of the variables. Temperature at 2m is more representative of the large scale than wind speed at 10m, which is more local. Given the resolution of the model, the wind scores are logically lower. Globally, it is noticeable that for meteorological variables, the nudging configurations have always better statistical scores, logically these variables being direcrly nudged. The offline configuration gives the best results, even if differences between online and offline are low.

- Table 2 and associated discussion: Please clearly define and distinguish Rs and Ra.

  The exact definition of these two statistical scores was added in the manuscript (and also here in the reply to Reviewer #1).

- Line 203: Which time period(s) does the statement beginning with "There is no evidence" refer to?

  As for all comments in this section, it is for the whole modelled period, i.e the two summer months of July and August. The sentence was modified such as:

> Finally, for the whole modelled period, there is no evidence as to which simulation best reproduces the observations, but the statistical scores ( Table **??** ) show that the simulations with nudging performs better.

- Lines 229 - 231: The results presented later in Table 4 which clearly show a suppressed coupling effect when nudging is employed seem to contradict the conclusion that spectral nudging does not interfere with the effects of aerosol-meteorology coupling. Are the conclusions presented here only applicable to the specific time period and handful of locations analyzed in Section 3.3? If so, a strong caveat to that effect is needed. In addition, it would be interesting to perform this analysis for the locations and time periods where the coupling effect is strongest in the no-nudging case, to quantify the dampening impacts of nudging when and where aerosol feedback effects are most pronounced. It might also help to stratify the analysis by time of day.

  The locations presented are just examples, as not everything can be shown in an article. But the statistics have been made for all the stations, and this shows that the stations chosen are representative.
  Both effects are important, and we can't conclude that using nudging makes it unnecessary to take coupling into account. We don't think that the impacts are very marked temporally, either on the period (daily or weekly variability) or on the time of day, as this would be very marked on hourly time series.
  The impact of coupling will be highly dependent on aerosol abundance, altitude, type and size distribution. The impact of nudging will be on a larger scale and at higher altitudes. This makes it difficult to isolate a time and place for analysis, and tends towards the particular case. We prefer to present things in terms of statistics and distributions. The aim is to give the amplitude of each impact so that the reader realizes that nudging is not a small effect compared to coupling.

- Lines 248 - 250. Can you provide a hypothesis or explanation for how nudging impacts aerosol size distributions in this case?

  Yes. The nudging will help the regional model to have a better large scale wind speed. The mineral dust scheme is the one of (Alfaro and Gomes, 2001). This scheme is wind-speed dependent for the dust emission, both for the intensity of the emitted flux and for the size distribution. By changing the wind speed, the size distribution of dust is changed then for the whole amount of aerosols. An explanation was added in the manuscript.

- Lines 273 - 275: Can you provide a hypothesis or explanation for these patterns?

  These patterns correspond to the mean averaged values over the whole modelled period. It is thus difficult to interpret nor to link them to surface characteristics, since meteorological perturbations (or changes) and pollutant plumes underwent numerous and very different horizontal and vertical transports over the entire period. This mode of representation is used here to give a general order of magnitude for impacts, and to compare the impact of processes that would otherwise be difficult to compare.

- Lines 276 - 282 / Figure 8: Please show and discuss all four cases for ozone, just like for water vapor mixing ratio Figure 7

  OK. The Figure 8 was changed and is now:

  We did not make this choice in the original manuscript because we did not see the added value. But, as it is a Reviewer request we added it and improved the discussion. The paragraph was updated and is now:

[Figure]

Figure 1: *Differences of surface ozone concentrations, time averaged over the period $1^{st}$:$31^{st}$ August 2022.*

The same difference calculations are done for surface ozone concentrations, Figure 1 . As for the water vapor, the differences are more significant for the impact of the nudging. The spatial structure are not directly comparable between the two variables. This is normal, as we are representing a surface quantity only, which is a secondary pollutant, potentially produced and transported in a completely different way to water vapor (presented vertically integrated). For the effects of the coupling, the differences are more significant and positive over North Africa with a maximum of $+3$ $\mu$g.m$^{-3}$. Over Western Europe the differences alternate between negative and positive values, but never exceed $\pm$ 1 $\mu$g.m$^{-3}$. The non-zero differences are spatially very limited and the majority of the differences are below the low value of $\pm$ 0.4 $\mu$g.m$^{-3}$. Figure 1 for nudg_online also shows much larger differences over the whole simulation domain. Positive and negative differences can be over sea or over land, no specific patterns are visible. Depending on the location and averaged over a month, the differences due to nudging can reach $\pm$ 6 $\mu$g.m$^{-3}$ for surface ozone concentrations.

- Figure 9 and Table 4: in addition to visualizing and summarizing these time-averaged data points, it might be interesting to prepare a scatter plot of (nonudg_online - nonudg_offline) on the x-axis and (nudg_online - nonudg_online) on the y-axis, with each datapoint in the scatter plot representing one specific hour and grid cell. Such a plot may reveal whether the dampening effect of nudging is potentially more or less pronounced depending on the strength of the undamped coupling effect.

  a revoir This kind of scatter-plot was tried but with too much points, it was unreadable. The domain

size is 103 × 106 × 20 grid cells, multiplied to 24h per day and 75 days of simulation (including the spin-up period), corresponds to 393 048 000 values. It is why all the results presented in this study correspond to time-averaged or surface values or distributions. The way in which the results are presented has been carefully thought through to ensure clarity and conciseness.

- Section 3.6: Aside from some discussion of latitudinal gradients at the end of this section, it is not clear why a latitudinal cross-section was chosen and why it was set up at this particular longitude. The discussion of vertical differences in interesting, but since this analysis is limited to one specific longitude it is unclear how representative these differences are across the domain. It is also not clear how to relate the results for the 3-day period discussed in this section to the rest of the modeling period and the longer-term averages analyzed in the previous sections. An additional way to present results for upper layers across a wider range of conditions might be to calculate vertical profiles averaged over all horizontal grid cells, along with their standard deviations for each layer.

  The choice for the specific longitude is better explained now:

  > Values are displayed along the latitude (from 15 to 55 ºN) and for the iso-longitude value of 5 ºE. This longitude correspond to the middle of the France and the place where the fire plumes passed over the Landes (where emissions were) and then Belgium and Germany (after transport).

  It is right that the time-average duration is different but it was to propose another type of view on the results focusing more on one type of high altitude transport event. For the proposition of an spatial average of all vertical profile, the risk is to have a non realistic signal, viewing the large spatial variability. In fact, there is an infinity of representation type when we are analyzing a three-dimensional two-month simulation. We produced a large number of figures, and the final choice was the one presented in the manuscript, as it best represents the diversity of situations encountered.

- Line 310: I am unclear about the meaning of "The most important changes are in altitude"

  OK, the sentence was rewritten as:

  > For the impact of the coupling, changes are more important in case of no spectral nudging. The largest differences are between 5000 and 8000 m with changes approximatively between ±1.5 ºC.

- Line 331: To help interpret the results for mineral dust emissions, could you please summarize the approach for calculating these emissions and their dependence on meteorology (e.g. wind speed, ambient and/or soil moisture, etc.)?

  Yes, OK. A paragraph was added in the 2.2 section (CHIMERE model configuration).

  > The mineral dust emissions are parameterized following (Alfaro and Gomes, 2001) and modified following (Menut et al., 2005). Vertical fluxes of emission is calculated such as the size distribution of the emission depends on the magnitude of the friction velocity, the soil distribution and its mineral characteristics. The humidity is taken into account via the soil moisture with the Fecan et al. (1999) parameterization. Precipitation and soil recovery for emission is also taken into account following (Mailler et al., 2017).

- Lines 336 - 343: Please add some discussion of the mechanisms and processes linking the temperature, wind speed, emissions, and AOD difference patterns shown in Figure 13.

  Add just a few lines to explain the link between temperature, wind, emissions and AOD is challenging being the whole geophysics. More precisely, for this specific case, some additional comments could be formulated and the following paragraph was added in the manuscript.

For temperature, the differences are both negative and positive and can reach $\pm$ 1.5 K. For the 10m wind speed, these differences are mainly negative (a reduction in wind) except over the sea, where local positive maxima can reach 2 m.s$^{-1}$. Due to the geophysics equations, there is no reason to have a direct link between temperature and wind speed at the surface when the nudging is used: the differences are not due to the geophysics, but is due to the forcing exerted by the large scale on the regional scale by two different models. For mineral dust emissions, the differences are localised where these emissions occur, i.e. mainly in North Africa. The main trend is negative differences showing that, on average, nudging tends to reduce these emissions. The differences in AOD represent a synthesis of the previous differences, this variable representing the aerosol load in the atmosphere and therefore reflecting changes in temperature and wind speed, and therefore dust emissions, their concentrations and therefore their optical thickness. There are wide spatial variations in AOD, with large positive structures over Africa, but also large negative structures over the south-western part of the domain, including a maritime area. The differences are important and around $\pm$ 0.15. For the large area in the south-west of the domain where AOD is lower, this could be mostly due to the also negative difference in the 10m wind speed. The AOD being representative of the whole atmospheric column and the 10m wind speed only representative of the surface, the link between the differences for these two variables can't be established any further than that.

**3.3   Technical Corrections**

All following corrections were taken into account. Thanks to the reviewer.

- Line 4: "processes" instead of "process"
- Line 12: "model" instead of "modeling"

  We really want to talk about 'modeling'.

- Line 14: suggest inserting "for this purpose" after "is used"  The sentence was already changed, after Reviewer #1 suggestion.

- Line 18: suggest rephrasing as "On the other hand, for chemistry-transport modeling (CTM) in online mode"
- Line 51: remove comma after "Using the WRF model"
- Line 61: change "that" to "than"
- Line 68 - 69 and elsewhere in the manuscript: change "pollutants concentrations" to "pollutant concentrations"
- Line 71: suggest changing "interplay" to "interact"
- Line 72: suggest changing "then it is important" to "making it important"
- Lines 73 - 75: remove "the" before "Section"
- Lines 82-83: suggest rewording to "The model was configured with and without spectral nudging and with and without taking into account aerosol direct and indirect effects".
- Line 133: suggest replacing "is to date the last distributed one" with "currently is the latest distributed one"
- Lines 171 - 172: move "ozone, PM2.5, and PM10" after "the three modeled chemical components"
- Line 175: "is less important and the scores are more or less the less" - this is unclear
- Line 198: change "contrarily" to "contrary" or "in contrast to"
- Line 199: please clarify "at the interface"
- Line 207: suggest changing "precise" with "detailed"
- Line 210: change "mi-August" to "mid-August"
- Line 238: change "that for the surface" to "as for the surface"
- Line 249: change "simulation" at the end of the line to "simulations"

- Lines 269 - 270: please avoid double occurrence of "particularly"
- Line 272: change "may negative" to "may be negative"
- Line 328: suggest changing "online or offline" to "online vs. offline"
- Line 371: change "lies on the fact" to "lies in the fact"
- Lines 373 - 374: suggest rewording "not only because emissions change but also because of feedbacks of aerosols on meteorology"

**References**

Alfaro, S. C. and Gomes, L.: Modeling mineral aerosol production by wind erosion: Emission intensities and aerosol size distribution in source areas, J. Geophys. Res., 106, 18,075–18,084, 2001.

Bessagnet, B., Menut, L., Lapere, R., Couvidat, F., Jaffrezo, J.-L., Mailler, S., Favez, O., Pennel, R., and Siour, G.: High Resolution Chemistry Transport Modeling with the On-Line CHIMERE-WRF Model over the French Alps-Analysis of a Feedback of Surface Particulate Matter Concentrations on Mountainous Meteorology, Atmosphere, 11, doi: 10.3390/atmos11060565, 2020.

Briant, R., Tuccella, P., Deroubaix, A., Khvorostyanov, D., Menut, L., Mailler, S., and Turquety, S.: Aerosol–radiation interaction modelling using online coupling between the WRF 3.7.1 meteorological model and the CHIMERE 2016 chemistry-transport model, through the OASIS3-MCT coupler, Geoscientific Model Development, 10, 927–944, doi: 10.5194/gmd-10-927-2017, 2017.

Fecan, F., Marticorena, B., and Bergametti, G.: Parameterization of the increase of aeolian erosion threshold wind friction velocity due to soil moisture for arid and semi-arid areas, Annals of Geophysics, 17, 149–157, 1999.

Kruse, C. G., Bacmeister, J. T., Zarzycki, C. M., Larson, V. E., and Thayer-Calder, K.: Do Nudging Tendencies Depend on the Nudging Timescale Chosen in Atmospheric Models?, Journal of Advances in Modeling Earth Systems, 14, e2022MS003 024, doi: https://doi.org/10.1029/2022MS003024, e2022MS003024 2022MS003024, 2022.

Mailler, S., Menut, L., Khvorostyanov, D., Valari, M., Couvidat, F., Siour, G., Turquety, S., Briant, R., Tuccella, P., Bessagnet, B., Colette, A., Létinois, L., Markakis, K., and Meleux, F.: CHIMERE-2017: from urban to hemispheric chemistry-transport modeling, Geoscientific Model Development, 10, 2397–2423, doi: 10.5194/gmd-10-2397-2017, 2017.

Menut, L., C.Schmechtig, and B.Marticorena: Sensitivity of the sandblasting fluxes calculations to the soil size distribution accuracy, Journal of Atmospheric and Oceanic Technology, 22, 1875–1884, 2005.

Menut, L., Cholakian, A., Siour, G., Lapere, R., Pennel, R., Mailler, S., and Bessagnet, B.: Impact of Landes forest fires on air quality in France during the 2022 summer, Atmospheric Chemistry and Physics, 23, 7281–7296, doi: 10.5194/acp-23-7281-2023, 2023.

Tuccella, P., Menut, L., Briant, R., Deroubaix, A., Khvorostyanov, D., Mailler, S., Siour, G., and Turquety, S.: Implementation of Aerosol-Cloud Interaction within WRF-CHIMERE Online Coupled Model: Evaluation and Investigation of the Indirect Radiative Effect from Anthropogenic Emission Reduction on the Benelux Union, Atmosphere, 10, doi: 10.3390/atmos10010020, 2019.

von Storch, H. and Zwiers, F.: Statistical Analysis in Climate Research, Cambridge University Press, 2001.